# Quantifying thermal adaptation of soil microbial respiration

Charlotte J. Alster [1,2] ✉, Allycia van de Laar[1,3], Jordan P. Goodrich[1,4], Vickery L. Arcus [1], Julie R. Deslippe [5], Alexis J. Marshall[1] & Louis A. Schipper[1]

Quantifying the rate of thermal adaptation of soil microbial respiration is essential in determining potential for carbon cycle feedbacks under a warming climate. Uncertainty surrounding this topic stems in part from persistent methodological issues and difficulties isolating the interacting effects of changes in microbial community responses from changes in soil carbon availability. Here, we constructed a series of temperature response curves of microbial respiration (given unlimited substrate) using soils sampled from around New Zealand, including from a natural geothermal gradient, as a proxy for global warming. We estimated the temperature optima ($T_{opt}$) and inflection point ($T_{inf}$) of each curve and found that adaptation of microbial respiration occurred at a rate of 0.29 °C ± 0.04 1SE for $T_{opt}$ and 0.27 °C ± 0.05 1SE for $T_{inf}$ per degree of warming. Our results bolster previous findings indicating thermal adaptation is demonstrably offset from warming, and may help quantifying the potential for both limitation and acceleration of soil C losses depending on specific soil temperatures.

Soils contain the largest reservoir of carbon in the terrestrial biosphere[1,2]. This store of carbon is broken down by soil microbes and released to the atmosphere through heterotrophic respiration[3] at a rate roughly ten times larger than the carbon dioxide ($CO_2$) emitted from anthropogenic sources[4]. Climate warming is expected to accelerate heterotrophic soil microbial respiration, greatly increasing carbon losses from soil to the atmosphere[5–7]. However, large uncertainty remains surrounding rates of soil carbon loss[6,8–10], including how microbial communities will adapt to higher temperatures[7,11,12]. Thermal adaptation of soil microbial respiration is central to understanding the resilience of natural systems to climate warming since adaptation of microbial temperature responses may further accelerate, or dampen, carbon cycle-climate feedbacks[6,13]. We note that it is the soil microbial communities that adapt to warming, with changes in the temperature response of respiration being an expected outcome of this adaptation; we refer to this here as the thermal adaptation of soil microbial respiration. Soil microbial communities may thermally adapt to warming through shifts in community composition or physiological adaptations[13,14]. While the literature in this area is vast, contradictory findings obscure whether soil microbial respiration will adapt to warming (e.g., refs. 11,15–17). Recently important progress toward resolving this conundrum has been made by measuring potential respiration rates across climate gradients (e.g., refs. 7,12,18). However, we still lack the ability to make predictions about soil microbial respiration as the climate warms, in part because we lack the continuous data to do so. Thus, the central question remains—does soil microbial respiration adapt to warming, and if so, by how much?

Several factors contribute to the inconsistencies found in prior work on thermal adaptation of the temperature response of soil microbial respiration. First, there are persistent methodological issues in the measurement and characterisation of temperature responses of respiration in soil systems[19–22]. In brief, the temperature response is typically measured over a narrow temperature range and modelled with disregard for biological thermal optima[19]. Persistent use of the $Q_{10}$

[1]Te Aka Mātuatua School of Science, The University of Waikato, Hamilton 3240, Aotearoa New Zealand. [2]Department of Soil & Physical Sciences, Faculty of Agricultural & Life Sciences, Lincoln University, Lincoln 7647, Aotearoa New Zealand. [3]Manaaki Whenua—LandcareResearch, Hamilton 3216, Aotearoa New Zealand. [4]Ministry for the Environment, Wellington 6143, Aotearoa New Zealand. [5]School of Biological Sciences, Victoria University of Wellington, Wellington 6012, Aotearoa New Zealand. ✉e-mail: charlotte.alster@lincoln.ac.nz

temperature coefficient, for example, leads to an oversimplification of the microbial temperature response as binary (increasing or decreasing). These issues are problematic because when characterisation of the initial temperature response is poor, comparing results across treatments, sites, and timescales for understanding thermal adaptation becomes futile. Confusion also remains regarding the relative versus absolute temperature sensitivity (Methods), which contributes to discrepancies in the literature[19,20,23]. Second, it is difficult to isolate the interacting effects of changes in microbial community responses from changes in soil carbon availability with warming[12,24]. As warming increases carbon limitation may also increase, making it difficult to evaluate changes in the intrinsic temperature response of soil microbial respiration from apparent changes[23,24]. Third, long-term warming studies are often accompanied by methodological challenges and artefacts that are often difficult to overcome[25,26]. For example, soil drying and changes in vegetation accompanying the warming treatment[26] can also influence microbial adaptation[27–32]. Most warming experiments are also limited to two treatments (warmed and control), which means that isolating a metric for the size of adaptation with increasing temperature is not possible. This lack of power, in both defining the temperature response and in selecting enough sites or treatments to discern changes, also provides a challenge and contributes to the uncertainty[19,33,34]. Consequently, while many studies predict an increase, decrease, or no change in microbial respiration with warming, few provide a quantitative metric for characterising the rate of this change.

Long-term soil geothermal gradients in Aotearoa New Zealand offer a unique opportunity to quantify rates of thermal adaptation across relatively controlled environmental conditions. Here, we constructed a series of temperature response curves ($n = 47$) from soils sampled along a natural geothermal gradient and from a range of soil temperatures across New Zealand (Fig. S1) as a proxy for global warming. Mean annual soil temperatures from these sites ranged from 11–35 °C, which is representative of many temperate ecosystems[35], making New Zealand an ideal case study for understanding soil microbial thermal adaptation to climate warming. To construct the temperature response curves, we incubated each of these soils with and without added glucose at >11 discrete temperatures ranging from ~4–42 °C using a temperature gradient block and then measured $CO_2$ flux (Fig. 1; Methods). The $CO_2$ flux from the added glucose fraction was mathematically separated from the soil organic matter (SOM) fraction to determine the temperature response of respiration by the

microbial community independently of carbon availability[36] (Methods).

We fit a modified version of macromolecular rate theory (MMRT) to each temperature response curve to estimate the temperature optima ($T_{opt}$) and inflection point ($T_{inf}$) as indicators of temperature sensitivity (Methods). Employing this approach, we were able to eliminate the confounding effects of measuring changes in temperature sensitivity across a narrow temperature range. We then estimated the rate of adaptation of $T_{opt}$ and $T_{inf}$ for microbial respiration across the environmental temperature gradient, accounting for spatial autocorrelation, and compared these values to differences in microbial community composition along the geothermal gradient (Methods). We hypothesised that as mean annual environmental temperature (MET) increases, the $T_{opt}$ and $T_{inf}$ of soil microbial respiration will also increase because microbial communities will thermally adapt to warmer soil conditions. By explicitly quantifying the rate of thermal adaptation in this study, we were able to make predictions about how thermal adaptation may alter rates of soil microbial respiration with climate warming.

## Results and discussion
### Thermal adaptation of soil microbial respiration
We found clear evidence for thermal adaptation of soil microbial respiration when measurements were not confounded by varying substrate availability (Fig. 2). Adaptation of the microbial respiration occurred at a rate of 0.29 and 0.27 degrees for $T_{opt}$ and $T_{inf}$, respectively, per degree of warming with a high level of confidence (±0.04 1SE for $T_{opt}$ and ±0.05 1SE for $T_{inf}$, $n = 47$; Fig. 2) when accounting for spatial relatedness. The relationships between $T_{opt}$ and $T_{inf}$ with MET were highly constrained despite the wide variety of climatic and edaphic properties of the sites sampled, demonstrating the general importance of long-term environmental temperature in determining the temperature response. Our results suggest that thermal adaptation of soil microbial respiration is occurring at a rate disparate from the rate at which global temperatures are warming (i.e., the rate changes in $T_{opt}$ and $T_{inf}$ is not 1 to 1 per degree of environmental warming).

While the vast majority of studies have been unable to explicitly quantify rates of thermal adaptation for soil microbial respiration in the absence of substrate limitation, there is some supporting evidence for slower thermal adaptation than the rate of warming, particularly for microbial growth rates. For example, Rinnan et al.[37] and Rijkers et al.[38] observed an increase in $T_{opt}$ of 0.07 to 0.27 °C per 1 °C increase in MET for soil bacterial growth from polar ecosystems. Others have observed an increase in $T_{min}$ (see Methods for definition) of 0.19 to 0.8 °C per 1 °C increase in MET for bacterial and fungal growth in more temperate ecosystems[39–41], averaging around 0.3 °C per 1 °C increase across ecosystems[42]. Li et al.[43] also found an increase of approximately 0.2 °C per 1 °C increase in MET for $T_{min}$ for soil microbial respiration measured from soils around China. Additionally, a meta-analysis of microbial temperature sensitivity from a variety of ecosystems and processes found an increase in $T_{opt}$ and $T_{inf}$ of slightly under 0.5 °C per 1 °C increase in MET[44].

The subtle shift in the temperature response identified here may also explain why previous studies of thermal adaptation of soil microbial respiration have frequently reported null results (e.g., refs. 17,45–47). Since thermal adaptation occurs at less than 0.3 °C per 1 °C increase in MET, and because most published experiments were not designed to detect such small shifts, it is unsurprising that evidence of thermal adaptation has rarely been reported. Data generated from ecological experiments are often inherently noisy and warming experiments are typically measured with minor differences in temperature (less than two degrees) between the warmed and control treatments (e.g., refs. 11,33,48). These factors make the detection of a warming response technically challenging. For example, 2 °C of warming would only elicit a 0.58 °C shift in $T_{opt}$, making measurement

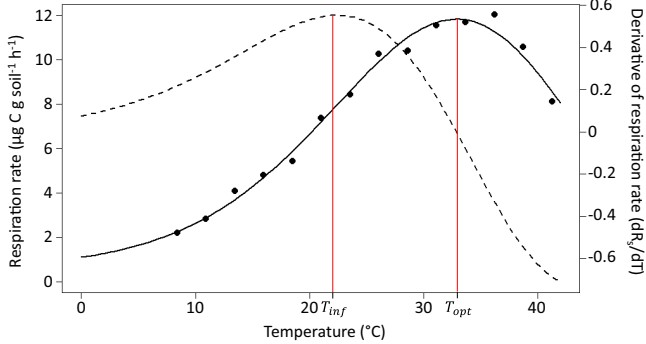

**Fig. 1 | Example temperature response curve.** Example temperature response curve for glucose-induced respiration from soil with a mean environmental temperature of 17.1 °C. Each black point represents the measured respiration rate from a different incubation temperature. The black, solid curve corresponds to the modified MMRT model fit and the dashed line corresponds to the first derivative of the respiration rate. The red lines mark the $T_{opt}$ (32.9 °C) and $T_{inf}$ (22.4 °C). The $T_{opt}$ is the peak of the MMRT curve and also where the derivative is zero. The $T_{inf}$ is the peak of the derivative and also where the slope is steepest on the MMRT curve.

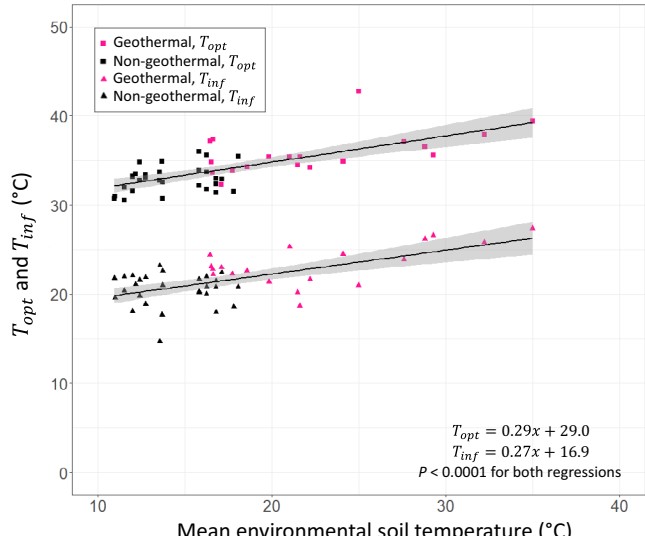

**Fig. 2 | Thermal adaptation of $T_{opt}$ and $T_{inf}$.** Regressions between mean environmental soil temperature and the $T_{opt}$ and $T_{inf}$ of soil microbial respiration of glucose. Square points indicate $T_{opt}$ values and triangle points indicate $T_{inf}$ values. Black points represent soils collected from geographically distributed sites across New Zealand and pink points represent soils collected from the geothermal gradient. For each regression (for $T_{opt}$ and $T_{inf}$), lines indicate the best fit with the grey, shaded regions indicating the 95% confidence intervals. Regressions account for spatial relatedness (Methods) and $x$ indicates mean environmental soil temperature in the equations.

of a significant change in the temperature response potentially difficult to detect. In the future, it would be prudent to design studies measuring the thermal adaptation of microbial communities with this in mind.

The mechanisms behind this incommensurate response of thermal adaptation of soil microbial respiration with warming remain elusive. Dispersal limitation is unlikely to have contributed to the offset in the warming response as the geothermal site has been active for decades[49] and the adapted communities were located within a few metres of each other. Additionally, microbial communities exhibit large genetic diversity and short generation times enabling their rapid evolution[50,51]. In contrast, a greater disconnect between thermal adaptation and the rate of warming is predicted for larger organisms[52–54], where evolutionary adaptation and migration rates are slower due to their longer lifespans. Perhaps the limitations to microbial thermal adaptation occur because the genomic regions encoding for respiratory proteins are highly conserved[55,56]. Respiration is a chemical pathway involving multiple enzymes that work in a coordinated manner[57]. Prentice et al.[58] hypothesised that the $T_{inf}$ of enzymes involved in metabolism are under strong selection pressure to maintain coordinated reaction rates across different environmental temperatures. If supported, the need to maintain coordinated activity across all respiratory enzymes could constrain thermal adaptation to warming, resulting in the offset of thermal adaptation and MET we observed.

Although the rate of thermal adaptation is relatively low, overall these results provide support for the optimum-driven hypothesis for thermal adaptation since $T_{opt}$ and $T_{inf}$ increase at approximately the same rate[19]. This suggests that the potential for soil C losses with warming may be buffered due to the shift in the temperature response curve, however, this buffering disappears as environmental temperatures approach the thermal optima (see Implications for climate change). The parallel shift in $T_{opt}$ and $T_{inf}$ deviates from what is predicted in the enzymatic literature, which finds slower thermal adaptation of $T_{inf}$ than of $T_{opt}$[59,60]. Given that we demonstrate that $T_{inf}$ shifts

more quickly than would be expected based on the thermodynamic properties of respiratory enzymes, our results suggest that other factors, such as the composition of the microbial community may contribute to the thermal adaptation of the temperature response. The differences observed between the enzymatic versus microbial community temperature response highlight the importance of testing hypotheses across scales since mechanisms can differ.

Notably, we also observed that the range of $T_{opt}$ values fell between 31 and 43 °C, which is an offset of 8–20 °C from the MET. While it is not entirely surprising that the $T_{opt}$ is higher than the MET[61–63], prevailing dogma posits that organismal evolution of enzymes should match their habitat[64,65]. The question then follows: why is the $T_{opt}$ so much greater than the mean environmental temperature? As was suggested earlier, underlying thermodynamic or environmental constraints may preclude perfect adaptation to environmental conditions due to fundamental physiological trade-offs[58,61,66]. Alternatively, perhaps it is not strategic for microbial respiration to evolve to match $T_{opt}$ to MET and a higher $T_{opt}$ provides some sort of biological safety under variable environmental temperatures[67,68]. It may be more advantageous for microbial adaptation of respiration to more closely match $T_{inf}$, as we see in this study, or coincide with maximum environmental temperature or reflect large temperature fluctuations instead[19,58]. Whatever the case, these results demonstrate that the position of $T_{opt}$ could be governed by factors additional to MET and provide further evidence that the thermal adaptation of microbial respiration is constrained.

### The role of the microbial community
The geothermal gradient provides an opportunity to explore the thermal adaptation of microbial respiration under long-term warming in the field, while minimising confounding factors (e.g., air temperature, vegetation, precipitation, soil type)[69]. Analysing the temperature response of soil microbial respiration from along the geothermal gradient independently of the other soils collected from around New Zealand to control for this variation, we found an increase of 0.22 and 0.23 degrees for $T_{opt}$ and $T_{inf}$, respectively, per degree of warming (±0.08 1SE for both for $T_{opt}$ and $T_{inf}$). These results corroborate the findings from the full dataset (Fig. 2). As the temperature range of the soil geothermal gradient was large while other differences were minimal, these data improve confidence in our results and diminish the likelihood that other factors within the full dataset bias our interpretation. Additionally, because the soils from around New Zealand consisted of distinct locations and soil types, we could assume these microbial communities differed[70,71]. We confirmed variation in microbial community structure along the geothermal by characterising bacterial DNA and lipids.

Despite only subtle shifts in the temperature response of microbial respiration, we found large variations in microbial community composition along the geothermal gradient (Fig. 3; Fig. S2), reflective of a shift in the relative dominance of heterotrophic to autotrophic taxa. Bacterial and archaeal richness and diversity significantly decreased with increasing MET (Fig. 3a–c; $P < 0.0001$), corroborating findings from Nottingham et al.[72] It is likely that these changes occurred due to decreases in pH along the geothermal gradient (correlation of $r = -0.91$ with MET; $P < 0.0001$) as pH is a well-known driver of microbial community composition[73,74]. Similarly, we observed changes in total bacterial and fungal biomass along the geothermal gradient (Fig. 3d; Fig. S2). Regardless of the driving mechanism, this information provides strong evidence that although microbial adaptation is occurring through changes in the microbial community along the geothermal gradient, corresponding changes to the temperature response of microbial respiration remain limited. This begs the question: why do only modest changes in the temperature response occur despite major changes in the microbial community?

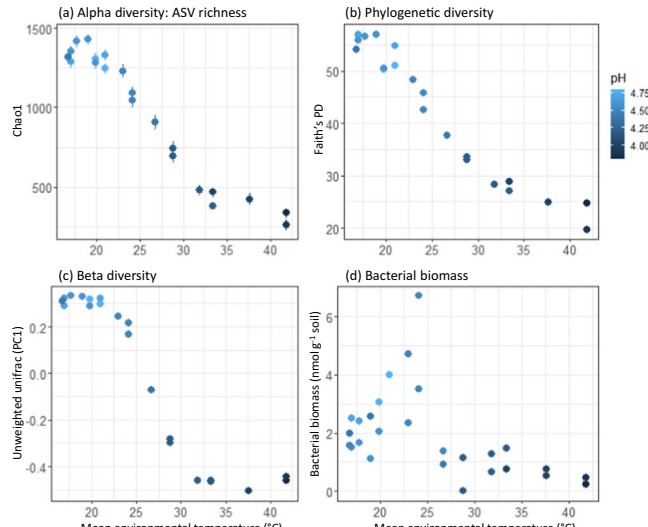

**Fig. 3 | Microbial community composition and abundance along the geothermal gradient. a** Alpha diversity of ASV richness (Chao1), **b** phylogenetic diversity, reported as Faith's Phylogenetic Diversity (PD), **c** beta diversity, estimated from the first principal coordinates axis (PC1) of the unweighted UniFrac analysis, and **d** total bacterial biomass. In all panels, points represent separate samples collected from along the geothermal gradient and are coloured by pH, which is strongly correlated with mean environmental temperature (Fig. S3). Data represented in panels (**a–c**) are from the 16S rRNA gene community analysis, while data from panel **d** represents the sum of bacterial Phospholipid Fatty Acids. Total microbial biomass (fungal + bacterial) can be found in Fig. S2.

The modest shift in the temperature response despite large changes in microbial diversity and composition point to high levels of functional redundancy (i.e., the ability of different microbial taxa to carry out the same functional process at similar rates regardless of composition[75,76]) among respiring microbes. This finding is consistent with the idea that specific features of the temperature response of soil microbial respiration are highly conserved among organisms and communities[55,56,58].

Alternatively, environmental temperature may not be the primary factor controlling the temperature response curve of soil respiration. While many factors were similar along the geothermal temperature gradient, notably pH and some soil nutrients did vary (Fig. S3; Tables S1 and S2). It is possible that these edaphic factors contribute to selecting which organisms adapt and survive along the temperature gradient[73,77], driving the selection of a subset of organisms with a specific type of temperature response. However, while pH is known to be a primary determinant of microbial community composition[73,74], and the microbial community composition may influence the temperature response of respiration[78], there is no direct mechanism by which pH would determine the temperature response of respiration. Prior studies have also failed to link pH and microbial temperature responses for respiration[44,79], however, it remains possible that an interaction between low pH and high temperature could cause stress to the microbe, indirectly affecting its temperature response. Interestingly, despite the fact that in our study pH is an important predictor of bacterial biomass, and possibly diversity, our models reveal that pH was not an important predictor of the temperature response (Table S3). Rather, the most parsimonious models of $T_{opt}$ and $T_{inf}$ included MET over pH (Table S3).

The increases in $T_{opt}$ and $T_{inf}$ we observed with warming could reflect changes in microbial community composition, as we hypothesised. Community composition may shift in order to match new environmental conditions; for example, we found thermophilic organisms like thermoplasmata occurring primarily at the warmest temperatures (Fig. S2; Tables S4 and S5). Alternatively, the increase in

$T_{opt}$ and $T_{inf}$ could simply reflect the observed decrease in prokaryotic diversity over the geothermal gradient, affecting the temperature response. In diverse soil microbial communities, the temperature response of respiration should reflect the sum of all single species' temperature response curves[80]. In simpler microbial communities, such as those we found in the warmest soil temperatures, the summation of the smaller number of species' individual temperature response curves may have resulted in differences in the measured temperature response parameters. The shift in temperature response within and between the different microbial communities may also reflect changes in the dominant glucose metabolising pathway along the gradient[81]. Large changes in microbial diversity and biomass, not accompanied by large changes in the temperature dependence of respiration, imply that respiration is a universal process whose temperature response is largely independent of community composition and abundance.

### Implications for climate change
Next, we explored what thermal adaptation of soil microbial respiration means in the context of climate change. We consolidated all of the temperature response curves into a single plot (Fig. 4a) to evaluate how thermal adaptation may alter respiration rates with warming. Current hypotheses about the thermal adaptation of soil microbial community are one dimensional in that only a single respiration rate is considered along with changes in MET[12,18,82]. By considering the full temperature response curve we are able to generate more dynamic hypotheses about how changes in MET affect respiration rates along a range of potential temperatures experienced in a given environment over the day and year.

To explain this concept in greater detail, we provide an example based on data presented in Fig. 4. If we have a soil with an annual temperature of 20 °C, on an average day (i.e., 20 °C), respiration would be roughly half of the total potential microbial respiration assuming substrate is not limiting (Fig. 4a). In contrast, on a warmer than average day (e.g., 32 °C) in the same soil with a MET of 20 °C, respiration would be closer to the maximum rate of potential respiration (Fig. 4a). For illustrative purposes, assuming that same soil warms 4.5 °C as is predicted by RCP 8.5 by the end of the twenty-first century[83], mean soil temperature in our example would increase to 24.5 °C. Without thermal adaptation, microbial respiration at 24.5 °C would increase by 18.1% compared to respiration rates pre-warming (Fig. S4). With thermal adaptation, there would only be a 14.1% increase in potential respiration at 24.5 °C (Fig. S4). However, on a warmer than average day (e.g., 32 °C), thermal adaption would cause 0.6% more potential respiration than was to be expected without thermal adaptation (Fig. 4b; Table S6). In contrast, on a cooler-than-average day (e.g., 17 °C), thermal adaptation would cause 7.0% less potential respiration than was to be expected without thermal adaptation (Fig. 4b; Table S6). This means that depending on the $T_{opt}$, $T_{inf}$, and instantaneous environmental temperature, the size of the change in soil microbial respiration with warming could be over or underestimated if thermal adaptation is not considered. These over and underestimations may be particularly accentuated during winter and summer months or day-night cycles.

Understanding of the temperature response curve at a range of different mean environmental temperatures for soil microbial respiration (Fig. 4a) also has other utilities. With knowledge of the MET, this approach could be used to estimate the percent of total potential respiration occurring at a given point in time at a particular site. Note, that it is potential respiration rate, because these experiments were conducted without substrate limitation and do not account for potential changes in substrate supply or microbial biomass, which may also adapt differentially to warming. Nonetheless, these values could be used for model parameterisation, for example by replacing $Q_{10}$ in relevant microbial and biogeochemical models[7,13,84], to

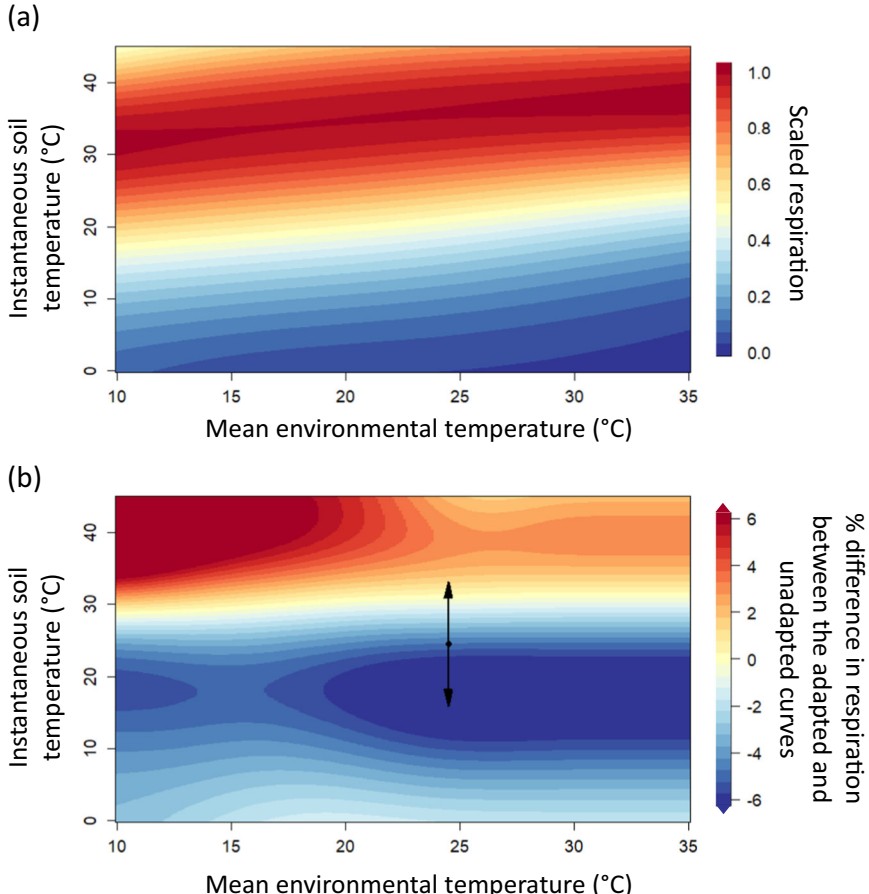

**Fig. 4 | Potential over and underestimation of soil microbial respiration without accounting for thermal adaptation under a climate change scenario.**
**a** Relationships between the mean environmental soil temperature and instantaneous soil temperature, assuming substrate is unlimited, based on our 47 temperature response curves. The dark red indicates the maximum potential rate of respiration (i.e., $T_{opt}$), while the dark blue indicates when respiration is zero.
**b** Percent difference in the total respiration rate between the thermally adapted and non-thermally adapted temperature response curves (i.e., the difference between panels **a** and **b** in Fig. S4) with 4.5 °C of warming. The black point and arrows represent an example with a MET of 24.5 °C and a higher-than-average temperature time point (32 °C) and a lower-than-average time point (17 °C). Differences between thermally adapted and non-thermally adapted potential respiration rates are also presented in Table S6.

account for changing temperature responses across temperature ranges[13,85].

Overall, our work demonstrates the importance of considering the entire temperature response curve when assessing the implications of thermal adaptation of soil microbial respiration on soil C loss. Depending on the specific mean and instantaneous soil temperatures, thermal adaptation of microbial respiration could either limit or accelerate potential soil C losses (Fig. 4b). This presents a theoretical expansion beyond the compensatory versus enhancement hypotheses leading to acceleration or dampening of C loss[12,18]. Here, we find that thermal adaptation could lead to both increased and decreased respiration from soil depending on the temperature range experienced by that soil. A large number of warmer-than-average days could potentially accelerate C losses from thermally adapted soil microbial communities. We should therefore be cautious about thinking of adaptation of microbial communities as either strictly compensating or enhancing soil microbial respiration; this binary way of thinking may also explain past empirical inconsistencies[47,86–88].

**Further considerations**
In this study, we characterised thermal adaptation for soil microbial respiration with high precision across a wide temperature range and removed the confounding factors associated with substrate availability. We found that the rate of adaptation was significantly lower than the rate of warming despite significant changes in the large and

diverse soil microbial communities, which are characteristics generally thought to increase the efficiency of adaptation and buffer against consequences of environmental change[50,51].

Although reasons for the offset of the temperature response of microbial respiration from warming remain unclear, the implications of our findings are important in the context of climate change. If the rate of adaptation were faster, it would lead to slower rates of respiration with thermal adaptation within most of the environmental temperature range (i.e., Fig. 4b would have a larger proportion of negative differences if the ratio of warming to adaptation was 1:1). Ultimately, this means greater potential for soil C losses compared to if rates of microbial adaptation were faster. Without considering the potential for thermal adaptation, we also risk underestimating potential respiration rates at temperatures above ~30 °C and overestimating potential respiration rates at temperatures at temperatures below ~30 °C (Fig. 4b). While this study lays the groundwork for understanding the temperature response of soil respiration, future work should aim to integrate substrate availability with thermal adaptation to better understand future soil C losses with warming, further understanding of changes in the temperature response of carbon use efficiency with warming, and investigate mechanisms behind the disconnect between thermal adaptation and warming. In summary, this research provides a rigorous quantification of the rate of thermal adaptation of soil microbial respiration by explicitly measuring full temperature response curves for a wide range of in situ soil

temperatures. Our results contribute to understanding how thermal adaptation may alter soil microbial respiration in a warming world.

## Methods

### Temperature sensitivity definitions and explanations

Multiple definitions of temperature sensitivity have long contributed to confusion in the discussion of the temperature dependence of soil microbial processes[19,20,42]. The $Q_{10}$ metric, which is the ratio of rates with a 10 °C increase in temperature, is most commonly used as an indicator of temperature sensitivity[7,19], despite the fact that $Q_{10}$ estimates only a relative temperature sensitivity (i.e., the temperature sensitivity changes based on the temperature measured). An alternative approach is to measure soil microbial respiration across a full range of biologically relevant temperatures as an indicator of absolute temperature sensitivity. By doing so we can capture the full temperature dependence of soil respiration, thus summarising all of the temperature sensitivities that can be attributed to a particular soil microbial process. The main benefit of this approach is that it makes comparisons among different soils, treatments, and studies more reliable[19], including comparisons for understanding thermal adaptation. While several different non-linear models may be appropriate to describe the temperature dependence of soil respiration[19,42,46], here we use a modified version of macromolecular rate theory (where heat capacity is temperature dependent[58,89,90]) to describe the temperature dependence of soil microbial respiration. The temperature sensitivity metrics used here to describe the temperature response curve are the temperature optimum and inflection point, which can also be estimated with other non-linear approaches[46,91].

Temperature optimum ($T_{opt}$): The $T_{opt}$ describes the point of maximum activity, which in this study corresponds to the maximum point of respiration.

Temperature inflection point ($T_{inf}$): The $T_{inf}$ describes the point of greatest change in respiration (i.e., point of maximum absolute temperature sensitivity) and has also been referred to as the point of maximum temperature sensitivity ($TS_{max}$)[44,55,78]. This point corresponds to the maximum of the first derivative (Fig. 1).

$T_{min}$ (the theoretical minimum for growth and activity): The $T_{min}$ is another commonly used metric to describe temperature sensitivity and corresponds to the theoretical minimum temperature for microbial growth and activity. $T_{min}$ is typically estimated using the square root (Ratkowsky) model[42] and has also been used as a proxy to describe thermal adaptation of soil microbial communities[42,43,67]. To estimate $T_{min}$ from the square root (Ratkowsky) model, values above $T_{opt}$ are removed for better estimation[42]. While $T_{min}$ cannot be calculated from MMRT since the x-intercept is not crossed, a positive relationship between $T_{min}$ and $T_{opt}$ has been observed[37,41,67]. However, this relationship is likely not 1 °C to 1 °C[37,41,67]. Even so, drawing comparisons between thermal adaptation for $T_{min}$, $T_{opt}$, and $T_{inf}$ may have some use for furthering our understanding of thermal adaptation. As $T_{min}$ is theoretical, in this study, we chose to focus on MMRT and parameters derived from MMRT (i.e., $T_{opt}$ and $T_{inf}$) in order to investigate how the entire temperature response curve adapts and influences C cycling.

### Study sites and soil sampling

The soil was collected at 48 locations, comprising 28 soils from farmland sampled across New Zealand and 20 samples from along a natural geothermal gradient located on the Arikikapakapa golf course in Rotorua, New Zealand (Fig. S1). These 48 soil samples spanned a mean annual temperature range of 11–35 °C.

The Arikikapakapa golf course is located in the Taupō Volcanic Zone, which has been geothermally active for thousands of years[92,93]. The soil geothermal gradient begins at a heated ground feature approximately 5.1 m long and 3.8 m wide, situated off the main golf course[94] (Fig. S5). Aerial photographs provide evidence that this feature has existed since at least 2003, but likely for the past several decades[49,95]. The soil consists of Tikitere siblings, which is an inactive hydrothermal recent soil of sandy loam texture[96]. Vegetation is dominated by *Axonopus affinis* and *Elymus repens* grasses, and unidentified mosses. The geothermal gradient and adjacent golf course are not fertilised or limed, but are frequently mowed.

Recorded temperatures along the gradient ranged from 63 to 5 °C and were continuously monitored using DS1922L iButton thermochrons (iButtonLink Technology). Five iButtons were buried at -5 cm depth and data collection occurred every 3-4 weeks beginning in August 2020. We observed a stable, long-term temperature gradient that changed both seasonally (highest in summer) and daily by about 5 °C. All of the rapid temperature variations observed near the geothermal source were linked to large rainfall events that occurred on the same day or the day before[97]. Temperature estimates for each of the soil samples were extrapolated from the mean annual values of the five iButtons using an exponential decay curve ($R^2 = 0.99$). The maximum and minimum average daily temperatures were also extrapolated from hourly means from the five iButtons. Soils selected for sampling from the geothermal gradient were randomly chosen to ensure roughly equal coverage of pre-determined thermal zones.

Elemental data from along the geothermal gradient was characterised using an Agilent 89,000 inductively coupled plasma mass spectrometry (ICP-MS) (Agilent Technologies, Santa Clara, California, United States). In brief, two soil samples were collected from 14 locations along the geothermal gradient and subsequently sieved, ground, and dried. Each sample (1 g) was acid digested with HCl and $HNO_3$ for 30 min at 80 °C. The samples were diluted with 100 mL of water and filtered through a 0.45 μM filter. The samples were diluted again to a 1 acid: 5 water solution. Ultrapure $HNO_3$ (200 μL) was then added to 10 mL of each diluted sample for the ICP-MS analysis. We also measured the pH for every sample. The pH correlated strongly with temperature along the geothermal gradient (Fig. S3).

Soils not collected from the geothermal gradient were collected by Manaaki Whenua−Landcare Research as part of a large, multi-year, nation-wide soil survey. In brief, to select the sites for the New Zealand soil survey a statistical analysis was conducted of historic soil organic carbon (SOC) data to estimate the sample size needed to detect changes in SOC across five land use classes. A sample size of 500 was selected to exceed the statistical requirements for analysis. Subsequently, sampling sites were determined using balance acceptance sampling, where sampling sites were selected to maximise the geographic distance between sites. These details are described in Hedley et al.[98]. A set number of sites in each land use class were selected for sampling each year, ensuring national representation each year, but there was no set sampling schedule within the year. These samples were placed on ice or refrigerated, and processed at the University of Waikato in the order in which they arrived. Due to the length of nation-wide soil sampling campaign and the intensity of the incubation measurements we conducted, not all of the samples collected by Manaaki Whenua−Landcare Research were included in this experiment. Once a sufficient sample size had been obtained for our analysis, as determined by a post-hoc power analysis (power >0.99), we ceased data collection for this experiment.

Soil classification information for each sample was obtained through S-map[96]. To estimate the annual soil temperature at -5 cm depth for each sampling location not located along the geothermal gradient, we used station data from New Zealand's National Institute for Water and Atmospheric Research (NIWA). We extracted NIWA climate monitoring network data (https://cliflo.niwa.co.nz/) using the clifro package in R[99] by locating the nearest operational station with at least 20 years of available daily soil temperature data. The mean of these daily soil temperatures was averaged from the past two years to use for our mean environmental temperature measurements for these

data. The maximum and minimum average daily temperatures were also obtained from this data.

All soil samples were collected over an approximately 12-month period (Table S1). Soils were sampled from the top 7.5 cm, however, methods of collection varied depending on soil type (Table S1). Once samples entered the laboratory, they were mixed and sieved to 2 mm to increase homogeneity for the temperature response measurements. Samples were either refrigerated at 10 °C until the temperature response measurements could occur or placed at room temperature if temperature response measurements would be occurring within one week of collection.

### Measuring the temperature response of respiration

To measure the temperature response of soil microbial respiration, we incubated soil samples in a temperature gradient block for 5 h. Soil (2 g) was weighed into 36–80 Hungate tubes (24 mL). We used two different temperature gradient blocks for this experiment, so the number of Hungate tubes depended on a number of factors, including the number of tubes that the temperature block could accommodate[36,100]. Each temperature gradient block was equipped with a heater on the end and a cooler on the other, and a solid aluminium block between with holes drilled to fit the Hungate tubes; however, the placement of the holes within the temperature blocks differed. One of the blocks consisted of three rows of 44 holes where control and treatment tubes were placed side-by-side in each row. This temperature block is described in detail in Robinson et al.[100] The other block consisted of 18 larger holes where the control and treatment tubes were placed in the same slot. This temperature block is described in detail in Numa et al.[36] The temperature blocks ranged from ~2 and 50 °C with 1–2 °C increments in each hole. Half of the Hungate tubes received 0.25 mL of distilled water (control) and the other half received 0.25 mL of a 450 mM glucose solution (treatment)[36]. These tubes were capped with rubber septa, sealed with aluminium crimps, and vortexed. The tubes (plus four blanks) were then incubated for 5 h in the temperature gradient block, with a pair of control and treatment tubes placed at the same temperatures within the gradient block. We repeated this setup for each of the 48 soils sampled.

After the 5-h incubations, the tubes were placed on ice and 1 mL of gas was removed from each headspace using an insulin syringe (Becton-Dickinson and co). The gas was then measured on an Infrared Gas Analyser (IRGA; LI-COR, LI-7000 $CO_2/H_2O$ Analyser) to determine the $CO_2$ concentrations[100]. To separate out the temperature response of non-substrate limited microbial respiration, we subtracted the $CO_2$-C of the control sample from the treatment sample at each measured temperature.

We chose this approach for measuring the temperature response of microbial respiration to eliminate the confounding effects of both differences in substrate availability between soil samples and changes in substrate availability with increasing temperature. Reaction rates in soils are governed by both abiotic (e.g., sorption/desorption) and biotic (i.e., enzyme-regulated activities) processes. Temperature affects both of these processes, but in different ways. Biological reactions are unimodal (i.e., have a temperature optimum), while non-biological reactions typically increase exponentially with increasing temperature[101]. If C becomes more available at higher temperatures due to increasing sorption/desorption processes (for example), then the measured temperature response would be conflated between these various biotic and abiotic mechanisms. There is also evidence to suggest that substrate availability in soils is a major factor influencing the temperature response of heterotrophic microbial respiration (e.g., ref. 102). By adding an excess of the readily available substrate and subtracting the control at each temperature, we eliminate the issue of more C becoming available at higher temperatures due to increasing abiotic processes and minimise potential discrepancies between soil samples when characterising the microbial temperature response.

To confirm that respiration rates increased linearly over time (and were not confounded by microbial growth at warmer temperatures), we also measured respiration from two soils (one from the geothermal gradient and one from an agricultural soil) every hour for 6 h at several temperatures with the added glucose. If microbial growth rates increased over the incubation period, we would have expected to see exponential increases in respiration. We found that respiration increased linearly with time (Fig. S6), minimising the likelihood that microbial growth during the short-term incubations significantly affected the respiration measurements.

### Estimating $T_{opt}$ and $T_{inf}$

To determine the temperature response, we used MMRT to fit the $CO_2$-C concentrations for the control and treatment samples with temperature:

$$\ln(R_s) = \ln\left(\frac{k_B T}{h}\right) - \frac{\Delta H_{T_0}^{\ddagger}}{RT} - \frac{\Delta C_P^{\ddagger}(T - T_0)}{RT} + \frac{\Delta S_{T_0}^{\ddagger}}{R} + \frac{\Delta C_P^{\ddagger}(\ln T - \ln T_0)}{R}$$

(1)

In this equation, $R_s$ is respiration rate, $k_B$ is Boltzmann's constant, $T$ is temperature (K), $h$ is Planck's constant, $R$ is the universal gas constant, $\Delta H_{T_0}^{\ddagger}$ (‡ superscript denotes transition state) is the change in enthalpy (J mol⁻¹), $\Delta S_{T_0}^{\ddagger}$ is the change in entropy (J mol⁻¹ K⁻¹), $\Delta C_P^{\ddagger}$ is the change heat capacity (J mol⁻¹ K⁻¹), and $T_0$ is the reference temperature (set to 300 K)[103]. We used a non-linear least-squares regression in R version 4.1.1[104] to fit MMRT and allowed $\Delta C_P^{\ddagger}$ to vary linearly with temperature:

$$\Delta C_P^{\ddagger} = A(T - T_0) + B$$

(2)

Here, the slope is $A$ and $B$ is the value of $\Delta C_P^{\ddagger}$ at $T_O$. We chose to vary $\Delta C_P^{\ddagger}$ linearly with temperature instead of holding it constant because $\Delta C_P^{\ddagger}$ is temperature dependent across this wide temperature range[58,89,90].

Because of anomalies observed for a few of the temperature response curves at the highest temperatures (i.e., a second increase in respiration rate following the initial decrease), we restricted model fitting to less than 42 °C for consistency, although data for all of the measured temperatures are available in the Table S7. A secondary increase in respiration rate at high temperatures could be due to the upregulation of metabolic pathways under thermal stress and has been observed previously[36,105,106]. However, investigating this phenomenon is beyond the scope of the current experiment. To compare model fits for varying $\Delta C_P^{\ddagger}$ versus maintaining a constant $\Delta C_P^{\ddagger}$ we used Akaike information criterion values corrected for a finite sample size (AICc). We found that varying $\Delta C_P^{\ddagger}$ was best or equivalent to keeping it constant for the majority of the soils (Table S8). Thus, we chose to use this version of the model to analyse our temperature response data. We also removed one of the samples from the analysis because the model would not converge during fitting, leaving us with a total of 47 temperature response curves in the final analysis (Fig. S7; Table S9). We estimated the $T_{opt}$ and $T_{inf}$ based on the MMRT curve fits, by identifying the largest predicted value and the largest difference between values generated along the modelled temperature response curves.

Next, we used Moran's $I$ to determine if $T_{opt}$ and $T_{inf}$ varied spatially ($P < 0.0001$). We then fit two spatial simultaneous autoregresion (SAR) models using the spatialreg package[107] with $T_{opt}$ and $T_{inf}$ as the dependent variables and MET as the independent variable to account for the location effect. SAR models incorporate spatial autocorrelation using a neighbourhood matrix specifying the relationship between the residuals at each site and between neighbours to analyse data with unequal spatial distributions. We also assessed if changes in pH could better explain $T_{opt}$ and $T_{inf}$ than MET using a series of linear regression models for both the geothermal gradient data separately and for the

full dataset. We checked models for both pH and MET independently and together, and compared them using AIC.

## Additional methodological analyses with $T_{opt}$ and $T_{inf}$

To check if seasonal variability or differences in sampling date influenced $T_{opt}$ or $T_{inf}$ we conducted linear regressions replacing the annual MET with the average temperature from 30-days prior to the sampling date for the geothermal gradient samples. We found no significant differences between the slopes of these regression lines (MET versus the 30-day temperature average) for either $T_{opt}$ or $T_{inf}$ ($P > 0.05$). For simplicity, we decided to use MET as the main environmental parameter analysed in this study since it is a common measure of environmental temperature and a relatively simple, gross measure of temperature at annual scales. Because soils were sampled at different dates and had slightly different storage periods, we also examined the effect of these on $T_{opt}$ and $T_{inf}$ and found no differences in either the sampling date or incubation date on $T_{opt}$ or $T_{inf}$ ($P > 0.05$). Lastly, we also conducted a t-test to confirm that $T_{opt}$ and $T_{inf}$ did not vary significantly between the two temperature blocks ($P > 0.05$).

## Scaling the temperature response curves

To combine all of the temperature response curves we scaled each curve individually so that respiration rates fell between zero and one, according to:

$$\text{scaled } \widehat{R}_s = \frac{\widehat{R}_s - \min(\widehat{R}_s)}{\max(\widehat{R}_s) - \min(\widehat{R}_s)} \qquad (3)$$

This allowed us to visualise the results from different soils together without changing any of the parameters associated with a given model fit. We generated the contour plot of scaled respiration rates as a function of environmental and instantaneous temperature using a cubic spline with three degrees of freedom to smooth the fitted curves over a continuous environmental temperature surface.

## Microbial community characterisation across the geothermal gradient

We assessed changes in the microbial community abundance and composition across the geothermal gradient using microbial phospholipid fatty acids (PLFA) and DNA analysis. For the microbial community analyses, soil cores (7.5 cm depth, 2.5 cm diameter) were taken from 14 points along the geothermal gradient (5, 10, 20, 30, 50, 80, 150, 200, 350, 500, 650, 1000, 1300, and 1600 cm from the geothermal feature). At each location, ~3 adjacent soil cores were taken from the left and right sides of the gradient ($n = 2$ at each sampling location) for a total of 28 samples. The corer was wiped with ethanol between each use. The sampling design here differed slightly from the sampling design for the respiration measurements. This was mainly to avoid oversampling at this unique site (less soil was needed for the microbial community characterisation) but still capture variability, and to avoid sampling from locations that were backfilled with other soil after sampling for the respiration measurements. After collection, soils were placed on ice in the field and then immediately mixed and sieved to 2 mm for consistency in the laboratory.

We used PLFA to estimate microbial biomass and community structure in each soil sample, which enabled the quantification of PLFA biomarkers for living microbial biomass[108] in µg fatty acid per g⁻¹ dry weight soil (µg g⁻¹ DW soil). Soils were collected for the PLFA analysis on the 31st of May 2021 and shipped on ice to Victoria University of Wellington for processing. Soil samples were lyophilised overnight (FreeZone 2.5 L Benchtop Freeze Dryer, Labconco, US) before ~0.5 g subsamples were weighed to 0.1 mg precision and subjected to the high-throughput PLFA method described by Lewe et al.[109] with slight modifications of the lipid extraction procedure. The low soil pH

required a doubling of the phosphate-buffer concentration used in the initial extraction stage. Lipids were extracted from freeze-dried soil samples using a chloroform:methanol:phosphate-buffer (1:2:1.6, v/v/v, pH 7.4). All other chemistry follows Lewe et al.[109]. In addition, samples were run in "split-less" mode during GC-MS analysis. We characterised 35 microbial PLFAs, designated according to standard nomenclature[110,111], and assigned these biomarkers to 6 microbial groups as per Table S10. Overall, we found total biomass to be relatively low in these samples, perhaps due to the acidic soil conditions[112,113].

To further analyse microbial community composition, we extracted total DNA from the soil samples using the PowerSoil DNA Isolation Kit (MoBio Laboratories, Carlsbad, CA, USA). Soils were collected for DNA sequencing on the 29th of March 2021. The sieved samples were frozen at −20 °C until DNA extraction could occur. DNA extraction, amplification, and sequencing all occurred at the University of Waikato.

We amplified the V4 region of the 16S rRNA gene in triplicate using the fusion-primer set 515F/806R[114]. For DNA extraction, we followed manufacturer instructions from the PowerSoil DNA Isolation Kit (MoBio Laboratories, Carlsbad, CA, USA) and quantified using a DeNovix DS-11 NanoDrop (Thermo Fisher Scientific). PCR reactions consisted of 20 µL reaction mixtures that included: 0.24 µM bovine serum albumin, 240 µM dNTP, 1.2x PCR buffer, 6 mM MgCl₂, 0.2 µM forward and reverse fusion primers, 1 U Platinum Taq polymerase (Thermo Fisher Scientific), and 2 ng of DNA. PCR thermocycling conditions were: initial 3 min at 97 °C for denaturation, followed by 27 cycles of 45 s at 94 °C, 1 min at 50 °C, and 1.5 min at 72 °C, and finally a 10 min incubation at 72 °C. The expected amplicon size of the PCR products was confirmed via electrophoresis with a 1% agarose TAE gel. Triplicate PCR products were pooled and normalised with SequalPrep™ (Thermo Fisher Scientific, United States) at an equimolar concentration into a single library for sequencing.

Amplicon sequencing was performed using the Ion PGM™ System for Next Generation Sequencing (Thermo Fisher Scientific, United States). Raw sequences were filtered with Ion PGM™ software to remove low-quality and polyclonal reads. For each sequence, the forward and reverse PCR primers were identified and trimmed with Cutadapt v2.3[115]. Those sequences without both primers were discarded. Amplicon Sequence Variants (ASVs) were processed using DADA2 v1.14.1[116] in R[104] with reads <230 bp, quality score <2, and expected error >2 removed. Taxonomy was assigned to ASVs with DECIPHER v2.22.0[117] using the SILVA v138 database[118]. Sequences were aligned using Multiple Alignment using Fast Fourier Transform (MAFFT) v7[119], and ASVs that were unclassified at the domain level or were Eukarya were removed from the analysis. A phylogenetic tree was generated using FastTree v2.1.11[120].

After sequence reads were quality filtered, seven samples were removed from the analysis since they had low sequence reads (<1200). Therefore, 21 samples were used in the final analysis. Singletons were also removed from each of the samples. Several diversity indices were used to assess differences in the microbial communities along the geothermal gradient. These analyses were conducted using the R packages phyloseq (version 1.42.0)[121], vegan (version 2.6.4)[122], and picante (version 1.8.2)[123]. We evaluated ASV richness using Chao1, phylogenetic diversity using Faith's PD, and beta diversity using the first principal coordinates axis of the unweighted UniFrac analysis. We ran a series of linear regressions to determine if MET or pH best explained each of these diversity and biomass measures and compared them using AIC.

## Reporting summary

Further information on research design is available in the Nature Portfolio Reporting Summary linked to this article.

## Data availability

The data generated in this study is available in the Supplementary Data and can also be accessed at: https://figshare.com/s/3746ef75599e608a5984. The raw DNA sequences presented in this study can be found at: https://www.ncbi.nlm.nih.gov/sra/PRJNA1002820 under the BioProject accession number PRJNA1002820.

## Code availability

All codes used in this study are openly available at *Zenodo*: https://doi.org/10.5281/zenodo.8248107.

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

## Acknowledgements

We wish to thank the groundskeeping team at the Arikikapakapa golf course for their continued help and support. We would also like to thank Manaaki Whenua—Landcare Research for sending us the soil samples from around New Zealand, particularly Georgie Glover-Clark for her help coordinating the sample shipments. We thank Roanna Richards-Babbage, John Longmore, Ian McDonald, and Craig Cary for their help with the sequencing and analysis, Andrew Barnes for his consultation on statistical measures, Emma Walker for providing a version of the MMRT code, Kristyn Numa for supplying preliminary data for methods validation, and Erland Bääth for his helpful comments on this manuscript draft. This work was supported by the Marsden Fund (Grant Number MFP-UOW1904; to V.A. and L.S.).

## Author contributions

V.A. and L.S. obtained funding. A.v.d.L. conducted the temperature monitoring and site characterisations, respiration measurements, and preliminary analysis of the data. J.D. performed the PLFA measurements and C.A. performed the DNA analysis with support from J.D. and A.M. V.A. provided the MMRT equation updates. J.G. scaled the temperature response curves and conducted the thermal adaptation analysis based on the RCP 8.5 predictions. C.A. conducted all other analyses and wrote the manuscript with significant input from L.S. All authors were involved with editing the manuscript and approved of the submitted version.

## Competing interests

The authors declare no competing interests.
