## [Peer Review File · Nature Communications]

Quantifying thermal adaptation of soil microbial respirationREVIEWER COMMENTS

Reviewer #1 (Remarks to the Author):

Review of NCOMMS-23-07213-T: "Quantifying thermal adaptation of soil microbial respiration" by Alster et al.

In this study, the authors team set out to investigate how environmental temperatures affect the temperature relationships for microbial decomposition (respiration rates). To accomplish this, the authors assemble a collection of soil samples, partly from a geothermal area with differences in soil temperatures, and partly based on a survey of soil samples in non-geothermal areas around New Zealand. The sampled soils were then assessed a range of temperatures (several positions along aluminium heat blocks cooled on one side, and heated on the other), plot the data, and fit a model to describe them. The authors are partial to the MMRT model, and use this to estimate T_{opt} and T_{inf} from the relationships. The authors note that T_{opt} and T_{inf} both increase by about 0.3 °C per 1°C higher environmental temperature, which they highlight as surprising. The authors also proceed to explain that such a response to climate warming will have complicated implications for rates of decomposition, depending on the instantaneous temperatures, and provide quantitative illustrations to make these points.

This is a well written and insightful account that provides suitable nuance and finesse to address a complicated topic. However, I have some questions and reservations about the approach, interpretation and claims that the results can support.

Substrate independent respiration?

Although this topic has been recently critiques and may not have any merit (Liang et al. 2023 GCB 29, 935-942), there is a persistent idea within the field of soil biogeochemistry and microbial ecology that the temperature sensitivity of decomposition (Q_{10} or T_{opt} , or any other metric) depends on soil C-quality. Specifically that "recalcitrant C" has a higher temperature sensitivity. These ideas have become accepted (e.g. Craine et al. 2010 Nat

Geosci) to the extent that their implications have been incorporated into how assessments of temperature effects are being made (e.g. Hartley et al. 2008; Bradford et al. 2008). One common recommendation is that, to avoid the potential bias that differences in C-quality can have on estimates of temperature effects, assessments should be made in ways that ensure substrate independence, for instance by supplying an abundance of substrate during short incubations (e.g. Bradford et al. 2008). The authors are well aware of this, and have performed their assessments of temperature effects on respiration on estimates of soil respiration in an abundance of glucose (l. 76, 346-351, etc). They then compare respiration with or without glucose and only study effects on the difference between the estimates (which is considered substrate independent). This seems like a clever approach, but I am worried that it also carries artefacts that will affect estimates of respiration, with an effect that grows with higher temperatures. I will try to explain, but this will require a detour.

One of the most well established methods to estimate microbial biomass in soil is the Substrate Induced Respiration (SIR) method. This relies on a very stable relationship between the amount of microbial biomass and the potential respiration (glucose saturated), as long as the time between glucose addition and respiration assessment is kept short (normally around 2h) that is robust between soils. The standard method for biomass was published by Anderson and Domsch (Soil Biol Biochem) in 1978, but it build on earlier work that had studied respiration rates resolved during longer timeframes (Anderson and Domsch, *Archiv für Mikrobiologie* volume 93, pages 113–127 (1973)). This earlier work showed that the addition of glucose capped respiration to a stable rates around 3-7-fold that in the unamended soil, which was stable for several hours (at 22°C, 3-12 h depending on soil and conditions). Following the stable plateau (which could be used to estimate microbial biomass, as described in 1978), respiration rates increased exponentially. More work revealed that CFU counts of microbes and other estimates of total microbial biomass followed a very similar patterns of increase, synchronized with the exponential respiration increases. The interpretation of these results was that glucose saturation had made respiration capped by the microbial community size, and when this started to increase (after the plateau), then the cap was released by a net increase in the microbial community size.

So what's the link? The authors use 5 h incubation times for their estimates of glucose

saturated soil respiration. However, these were performed at temperatures between 2°C and 50°C (l. 346). As the authors know well, microbial process rates accelerate with temperature. This is true for respiration, but also for growth rates. This means that the authors have effectively performed respiration rate estimates under glucose saturation that consumed very little of the added glucose, produced little respiration, and allowed for very little microbial growth at cold temperatures, and very much glucose consumption, very much respiration, and allowed for very much microbial growth at high temperatures. If we would convert these difference in temperature to what they would translate to in time at 22°C, a roughly 9 fold reduction in time at 2°C, and a 27-fold increase at 50°C. That is, with 5 h at room temperature as a reference (to match the Anderson and Domsch conditions, above), the 2°C incubation would be equivalent to a 33 min incubation at 22°C, and the 50°C equivalent to a 135 h incubation at 22°C (I am assuming a Q10 throughout the temperature range from 2 to 50°C of 3, which is of course not accurate; but better than assuming no temperature dependence). The equivalent to a short incubation is no conceptual problem (but may be hard to measure accurately), but the equivalent to a very long incubation (135 h!) brings about the issues that the community will start increasing their growth rate during the assessment. A good illustration is the challenge created can be found in Fig 1 in <https://onlinelibrary.wiley.com/doi/10.1111/j.1747-0765.2007.00235.x>. Consider the 4000 µg addition curve, and imagine include the initial 0.5 h compared to 135 h cumulative respiration estimated. The former represents the initial microbial community respiring at glucose saturation, but the latter is represents the microbes that have been induced to grow and accumulate during the equivalent to several days of glucose saturated growth, and have little to do with the initial community.

So how can these estimates of respiration at different temperatures be interpreted? I am really not sure. Especially problematic is that this effect is progressively growing in important with higher temperature, thus being perfectly confounded with what you are hoping to determine...

However, I do think there is hope! I think the data from the soil samples without glucose amendments is a more valuable indicator of the performance of the initial soil community (and that any bias due to C-quality is far smaller than that introducing by first growing a

large community on glucose, in the higher temperatures). I was surprised not to see a direct comparison between “substrate independent respiration” (the metric now used, in the authors current interpretation) and “basal soil respiration”. After all, if C-quality would confound the temperature assessment, a direct comparison between these could reveal that, which would be valuable and interesting to assess. And if the same pattern was found for both, this would be evidence against the CQH. Since the comparisons were not done, I would guess that they did not show the same patterns? Either way, the basal respiration assessments may hold the key...

Respiration is only a part of the microbial C-use

The authors central question is “does soil microbial respiration adapt to warming, and if so, by how much” (l. 44-45). However, to understand the soil-atmosphere C balance response to warming, this is only a partial assessment. It is not the “respiration” that adapts; it is the microbes. It is unlikely that only one of their processes would “adapt”. Hence, the full microbial-C-use is needed. After all, it is increasingly recognized that not only the pathway out of soil (respiration, but also the pathway of C into soil with long residence times is via microbes (The Microbial Carbon Pump, MCP, Liang, Schimel, Jastrow, Nature Microbiology, 2017). So, without estimates of the temperature relationships for microbial growth rates, how should we predict that these respond to differences in environmental temperatures? My best guess would be “the same as those for respiration”. If that it is the case, is this not then a status quo? When the microbial adaptation to temperature leads to more C respiration, it will simultaneously lead to more C used for growth (and potentially stored via the MCP)? Naturally, the jury is still out on this until estimates are available, but to disregard this part of the C-use equation of the microbes (that are the ones adapting to temperature!) seems an oversight. This will require reconsideration.

Are the findings new, or confirmatory?

The authors conclude with “... this research is the first to explicitly quantify the rate of

thermal adaptation of soil microbial respiration using a rigorous approach to measuring the full temperature response curve". Actually, by the authors own appropriate and scholarly literature comparison, this is not the case. Earlier compilations of published data (Bååth, 2018 GCB, MS reference list) and recent large scale surveys (Li et al. 2020; GCB; MS reference list) have both landed on similar estimates for how indices for temperature relationships depend on climate (0.2-0.3 °C change per 1°C warmer MET). Given this, it is awkward to call the findings "unexpected" (l. 307), and certainly to call them "the first" (l. 321). In fact, the estimates fall where they should, based on the few earlier studies that exist, making the finding confirmatory (but naturally still very valuable).

Link between taxonomic composition and temperature? Or pH?

The authors are happy to report a strong link between community diversity measures and environmental temperatures (and microbial temperature adaptations!), as seen in Fig. 3 a, b, c.

However, the authors also show a perfect correlation between pH and MET in their data on bacterial biomass (Fig. 3d). I wonder what would happen if the x-axes of Figs. 3a, b c would be changed from MET to pH? I think there would be a perfect correlation. While one interpretation of this could be that MET was the causal driver, the perfect confounding factor pH may be a more likely factor. Either way, walling (MET) over the other is not appropriate.

Other comments

l. 24 "support previous findings that indicate that..."

l.25. Acceleration of soil C losses... What about the microbial role in the formation of SOC vis growth?

l. 44-45. This disregards one of the two major fractions microbes use C for; respiration, but also growth (which could form SOC).

I. 74-77. I think the approach of using a 5 h incubation time at 2-50°C with a surplus of glucose introduces a systematic confounding effect. See general comments.

I. 88. Should changes be differences (they don't change over time; they vary in the landscape?)

I. 132. The problem is not actually "gradual" but that the effect is too small.

I. 137-140. Good point!

I. 142. Unexpected... But there were earlier reports (you cite them). See general comments

I. 150. I am not sure this rationale makes sense. You can find high levels, if MET are sufficiently high. This rationale could explain a cap in the level, but a small effect size per warmer temperature (throughout a wide range of MET)...

I. 169-182. This is interesting, but it requires a careful deconstruction of your used metric MET, and how it was obtained. Perhaps it would make sense to consider the maximal values of running means (24 h periods)? Should you consider seasonal change? The samples were measured over a long segment of time, which likely varied seasonally in temperature. How was this considered?

L. 203-204. Perhaps the most likely explanation is soil pH being confounded? This can also be explored with the taxa uncovered? Do the taxa change like they have been shown to along other pH gradients? See general comment

L. 214. You can evaluate how likely, based on the taxa and their relative abundances. pH has a very strong phylogenetic signal. Test it!

Fig. 3d. The estimates of bacterial PLFA concentrations seems low. Typical estimates are about 1000 nmol per g SOC. It looks like you are far lower here.

I. 246. Actually, the microbes adapt. A consequence of this can be a change of traits,

including respiration, but also of other things, like growth. If both change identically – status quo? At least it needs to be acknowledged.

I. 262. There WOULD only

I. 261-267. This is interesting, but stops short of a little extra sophistication. What about seasonal and diurnal variation in temperatures? These will similarly contribute (asymmetrically) to whether the adaptation will accelerate or slow C loss... What is MET, and why should a metric average be a useful predictor?

I. 302. See general comments

I. 328-337. Critical information is how estimates of MET were obtained, where they were measured, and how they were compiled. Did different storage time affect them (12 month period of collection)? What were the MET ranges? Seasonal max?

I.347-351. I think this introduces a powerful bias (and artefact) with temperature. See general comments

Reviewer #2 (Remarks to the Author):

Quantifying the thermal adaptation of soil microbial respiration is crucial to understanding the potential of carbon cycle feedbacks under climate warming. This manuscript entitled “Quantifying thermal adaptation of soil microbial respiration” by Alster et al. reports a series of temperature response curves of microbial respiration using soils from around New Zealand including from a natural geothermal gradient. The authors estimate the temperature optima (T_{opt}) and inflection point (T_{inf}) of each curve and document that adaptation of microbial respiration occurred at a rate of $0.29^{\circ}\text{C} \pm 0.04$ 1SE for T_{opt} and $0.27^{\circ}\text{C} \pm 0.05$ 1SE for T_{inf} per degree of warming. Finally, the authors conclude that thermal adaptation is demonstrably offset from warming, and calculate the potential for both limitation and acceleration of soil C losses depending on specific soil temperatures. Obviously, measurement of the rate of thermal

adaptation of soil microbial respiration is well valuable. Therefore, I think the results of this manuscript is very interesting and meaningful. However, I have several concerns about the experimental design and associated results:

1. It is hard to understand how the author combine soil samples from around New Zealand (large spatial scale) and samples from a natural geothermal gradient (small spatial scale) together to analyze excluding scale differences. Obviously, the two batch of samples are from two different ecosystems, farm land and recreation land. Furthermore, although the authors reported that all 48 soil samples were collected over an approximately 12-month period, all samples from farm land were collected in September to December and all samples from recreation land were collected more randomly in the 12-month period. By the way, the authors don't provide detailed distribution of these soil samples around New Zealand. Overall, the sampling strategy in this study lead to too many biases seriously weakening the reliability of the results.

2. Based on the description in the methods, the authors collected new 28 soil samples from 14 points along the geothermal gradient to perform microbial community analyses rather than using the same samples as measurement of the temperature response of soil microbial respiration. Why? The authors don't report the sampling time and specific locations. If these samples don't match the measurement of the temperature response of soil microbial respiration, it is not reasonable to explain thermal adaptation of soil microbial respiration by using the data of PLFA or DNA analysis in this study.

3. It seems the author estimate the relationship between mean environmental soil temperature and the T_{opt} and T_{inf} of soil microbial respiration by using annual mean environmental soil temperature. If so, the T_{opt} and T_{inf} of soil microbial respiration estimated by soil samples from different month can't represent the mean value across the whole year. It is not reasonable to quantify relationship between MET and the T_{opt} and T_{inf} .

4. In the result part about Thermal adaptation of soil microbial respiration, the authors perform analysis by using both samples from farm land around New Zealand and from the

natural geothermal temperature gradient. However, only samples from the geothermal gradient were further analysis in the role of the microbial community. Strictly, it is not acceptable to explain the overall thermal adaptation obtained in this study by only using a part of samples.

5. Line 122-123: the authors don't show any results about T_{min} in the study. I don't understand why the T_{min} should be discussed here.

6. Line 218-219: this sentence is not rigorous. Definitely temperature is important to control respiration rates, but the other factors, such as soil moisture, also strongly control respiration rates. Therefore, definite evidence should be provided in this study to document your viewpoint.

7. Please provide links for the raw data of amplicon sequences.

Below are our specific responses to reviewer comments (in bold). Note: Line numbers correspond to the manuscript version without track-changes.

Reviewer #1:

In this study, the authors team set out to investigate how environmental temperatures affect the temperature relationships for microbial decomposition (respiration rates). To accomplish this, the authors assemble a collection of soil samples, partly from a geothermal area with differences in soil temperatures, and partly based on a survey of soil samples in non-geothermal areas around New Zealand. The sampled soils were then assessed a range of temperatures (several positions along aluminium heat blocks cooled on one side, and heated on the other), plot the data, and fit a model to describe them. The authors are partial to the MMRT model, and use this to estimate T_{opt} and T_{inf} from the relationships. The authors note that T_{opt} and T_{inf} both increase by about $0.3\text{ }^{\circ}\text{C}$ per 1°C higher environmental temperature, which they highlight as surprising. The authors also proceed to explain that such a response to climate warming will have complicated implications for rates of decomposition, depending on the instantaneous temperatures, and provide quantitative illustrations to make these points.

This is a well written and insightful account that provides suitable nuance and finesse to address a complicated topic. However, I have some questions and reservations about the approach, interpretation and claims that the results can support.

Thank you for your thoughtful comments and interest in our manuscript. We have responded to these comments and updated the manuscript accordingly.

Substrate independent respiration?

Although this topic has been recently critiqued and may not have any merit (Liang et al. 2023 GCB 29, 935-942), there is a persistent idea within the field of soil biogeochemistry and microbial ecology that the temperature sensitivity of decomposition (Q_{10} or T_{opt} , or any other metric) depends on soil C-quality. Specifically that “recalcitrant C” has a higher temperature sensitivity. These ideas have become accepted (e.g. Craine et al. 2010 Nat Geosci) to the extent that their implications have been incorporated into how assessments of temperature effects are being made (e.g. Hartley et al. 2008; Bradford et al. 2008). One common recommendation is that, to avoid the potential bias that differences in C-quality can have on estimates of temperature effects, assessments should be made in ways that ensure substrate independence, for instance by supplying an abundance of substrate during short incubations (e.g. Bradford et al. 2008). The authors are well aware of this, and have performed their assessments of temperature effects on respiration on estimates of soil respiration in an abundance of glucose (l. 76, 346-351, etc). They then compare respiration with or without glucose and only study effects on the difference between the estimates (which is considered substrate independent). This seems like a clever approach, but I am worried that it also carries artefacts that will affect estimates of respiration, with an effect that grows with higher temperatures. I will try to explain, but this will require a detour.

One of the most well established methods to estimate microbial biomass in soil is the Substrate Induced Respiration (SIR) method. This relies on a very stable relationship between the amount of microbial biomass and the potential respiration (glucose saturated), as long as the time between glucose addition and respiration assessment is kept short (normally around 2h) that is robust between soils. The standard method for biomass was published by Anderson and Domsch (Soil Biol Biochem)

in 1978, but it build on earlier work that had studied respiration rates resolved during longer timeframes (Anderson and Domsch, *Archiv für Mikrobiologie* volume 93, pages 113–127 (1973)). This earlier work showed that the addition of glucose capped respiration to a stable rates around 3-7-fold that in the unamended soil, which was stable for several hours (at 22°C, 3-12 h depending on soil and conditions). Following the stable plateau (which could be used to estimate microbial biomass, as described in 1978), respiration rates increased exponentially. More work revealed that CFU counts of microbes and other estimates of total microbial biomass followed a very similar patterns of increase, synchronized with the exponential respiration increases. The interpretation of these results was that glucose saturation had made respiration capped by the microbial community size, and when this started to increase (after the plateau), then the cap was released by a net increase in the microbial community size.

So what's the link? The authors use 5 h incubation times for their estimates of glucose saturated soil respiration. However, these were performed at temperatures between 2°C and 50°C (l. 346). As the authors know well, microbial process rates accelerate with temperature. This is true for respiration, but also for growth rates. This means that the authors have effectively performed respiration rate estimates under glucose saturation that consumed very litte of the added glucose, produced little respiration, and allowed for very little microbial growth at cold temperatures, and very much glucose consumption, very much respiration, and allowed for very much microbial growth at high temperatures. If we would convert these difference in temperature to what they would translate to in time at 22°C, a roughly 9 fold reduction in time at 2°C, and a 27-fold increase at 50°C. That is, with 5 h at room temperature as a reference (to match the Anderson and Domsch conditions, above), the 2°C incubation would be equivalent to a 33 min incubation at 22°C, and the 50°C equivalent to a 135 h incubation at 22°C (I am assuming a Q10 throughout the temperature range from 2 to 50°C of 3, which is of course not accurate; but better than assuming no temperature dependence). The equivalent to a short incubation is no conceptual problem (but may be hard to measure accurately), but the equivalent to a very long incubation (135 h!) brings about the issues that the community will start increasing their growth rate during the assessment. A good illustration is the challenge created can be found in Fig 1 in <https://onlinelibrary.wiley.com/doi/10.1111/j.1747-0765.2007.00235.x> . Consider the 4000 µg addition curve, and imagine include the initial 0.5 h compared to 135 h cumulative respiration estimated. The former represents the initial microbial community respiring at glucose saturation, but the latter is represents the microbes that have been induced to grow and accumulate during the equivalent to several days of glucose saturated growth, and have little to do with the initial community.

So how can these estimates of respiration at different temperatures be interpreted? I am really not sure. Especially problematic is that this effect is progressively growing in important with higher temperature, thus being perfectly confounded with what you are hoping to determine...

Thank you for highlighting this important point and for your thorough explanation about your concern. We agree that potential acceleration of microbial growth at the different temperatures could result in bias that could be particularly confounded with the glucose additions as you point out with Anderson & Domsch 1973. Recognising this issue, we conducted preliminary experiments to check that respiration rates with glucose addition increased linearly over time at a range of temperatures. We regret not mentioning this in the original manuscript submission. We measured respiration rates of two soils (one from the geothermal gradient and one from an agricultural soil) with glucose addition every hour for 6 hours at several temperatures. If growth rates increased over the incubation period, we would have expected to see exponential increases in respiration like was observed in Fig. 1 of Sawada et al. 2008 (the paper you linked). Additionally, if glucose uptake was saturated at the warmest temperatures like was observed in the longer incubations in Anderson & Domsch 1973, then over time we would have expected

unimodal shaped curves at those high temperatures. However, respiration rates were linear with time regardless of the temperature ($R^2 > 0.88$) (figure below). This minimized concern that microbial growth or saturation of glucose uptake during the short-term incubations significantly affected the respiration measurements. We chose to measure at 5 hours to best capture respiration rates from samples with low activities. We added this figure, along with relevant explanatory text to Supplementary Information section 1.4 and to lines 356-359 of the manuscript.

Figure: Respiration measured over time for an agricultural and geothermal soil. Colour indicates incubation temperature and lines indicate best fit.

Additionally, it is also worth noting that there is growing evidence to suggest that generation times for soil microbes are quite slow. In a recent experiment using lipid stable isotope probing, generation/turnover times of *in situ* soil microbes ranged from ~20-60 days¹. Thus, while it is possible that the glucose addition spurred rapid growth of select, fast-growing microbial taxa confounding respiration rate measurements, since the majority of standing soil microbial biomass is slow-growing (yet still respiring), the potential for this bias remains limited.

1. Caro, T. A., McFarlin, J., Jech, S., Fierer, N. & Kopf, S. Hydrogen stable isotope probing of lipids demonstrates slow rates of microbial growth in soil. *Proc. Natl. Acad. Sci.* 120, 16 (2023).

However, I do think there is hope! I think the data from the soil samples without glucose amendments is a more valuable indicator of the performance of the initial soil community (and that any bias due to C-quality is far smaller than that introduced by first growing a large community on glucose, in the higher temperatures). I was surprised not to see a direct comparison between “substrate independent respiration” (the metric now used, in the authors current interpretation) and “basal soil respiration”. After all, if C-quality would confound the temperature assessment, a direct comparison between these could reveal that, which would be valuable and interesting to assess. And if the same pattern was found for both, this would be evidence against the CQH. Since the comparisons were not done, I

would guess that they did not show the same patterns? Either way, the basal respiration assessments may hold the key...

We appreciate you bringing up this excellent point regarding the carbon quality hypothesis (which as the reviewer points out has recently been suggested as being a mathematical artefact Liang et al 2023). However, our primary concern about using the soil samples without the glucose amendment was not differences in C *quality*, but rather differences in C *availability* (both between samples and at different temperatures). Reaction rates in soils are governed by both abiotic (e.g., sorption/desorption) and biotic (i.e., enzyme-regulated activities) processes. Temperature affects both of these processes, but in different ways. Biological reactions are unimodal (i.e., have a temperature optimum), while non-biological reactions typically increase exponentially with increasing temperature². If C becomes more available at higher temperatures due to increasing sorption/desorption processes (for example), then the measured temperature response would be conflated between these various biotic and abiotic mechanisms. There is also evidence to suggest that substrate availability in soils is a major factor influencing the temperature response of heterotrophic microbial respiration (e.g., ref³). By adding an excess of readily available substrate to measure the microbial temperature response, we eliminate the issue of more C becoming available at higher temperatures due to increased abiotic processes and minimize potential discrepancies between soil samples.

When we analysed the data from the non-glucose amended soil samples, we found evidence for exactly this issue. Several of the samples had a clear temperature optimum, albeit higher than the glucose-amended soils, but many of them increased log-linearly with temperature (see examples in the figure below). Therefore, to isolate the microbial temperature response we subtracted the “basal respiration” rates, which are a composite of the biological and non-biological reaction rates, from the glucose temperature response. We have improved description of our rationale and have clarified the text accordingly (lines 365-377).

Figure: Example temperature response curves of soil respiration without glucose amendment. The first example (FODAF) does not exhibit a clear T_{opt} , suggesting that abiotic processes may dominate the reaction rates in this soil. In contrast, the second example (MIJOZ) shows a clear T_{opt} , indicating a strong biotic contribution to respiration rates. This complicates the comparison of thermal adaptation rates between the microbial communities in these two soils using this approach, as the confounding factor of biotic versus abiotic influences is not accounted for.

2. Schipper, L. A. *et al.* Shifts in temperature response of soil respiration between adjacent irrigated and non-irrigated grazed pastures. *Agric. Ecosyst. Environ.* 285, 106620 (2019).

3. Moinet, G. Y. K. *et al.* The temperature sensitivity of soil organic matter decomposition is constrained by microbial access to substrates. *Soil Biol. Biochem.* 116, 333–339 (2018).

Respiration is only a part of the microbial C-use

The authors central question is “does soil microbial respiration adapt to warming, and if so, by how much” (l. 44-45). However, to understand the soil-atmosphere C balance response to warming, this is only a partial assessment. It is not the “respiration” that adapts; it is the microbes. It is unlikely that only one of their processes would “adapt”. Hence, the full microbial-C-use is needed. After all, it is increasingly recognized that not only the pathway out of soil (respiration, but also the pathway of C into soil with long residence times is via microbes (The Microbial Carbon Pump, MCP, Liang, Schimel, Jastrow, *Nature Microbiology*, 2017). So, without estimates of the temperature relationships for microbial growth rates, how should we predict that these respond to differences in environmental temperatures? My best guess would be “the same as those for respiration”. If that it is the case, is this not then a status quo? When the microbial adaptation to temperature leads to more C respiration, it will simultaneously lead to more C used for growth (and potentially stored via the MCP)? Naturally, the jury is still out on this until estimates are available, but to disregard this part of the C-use equation of the microbes (that are the ones adapting to temperature!) seems an oversight. This will require reconsideration.

Thank you for pointing out the nuance that it is the microbes who adapt, not respiration itself. We now make note in the first paragraph of the manuscript (lines 24-27).

Understanding the temperature response of microbial growth is certainly important, but beyond the scope of this study. While interconnected, the temperature response of microbial growth and respiration are distinct (e.g., refs^{4,5} and unpublished data – in prep). Microbial respiration controls CO₂ production rates from soils and is therefore the most direct link of soils to carbon cycle-climate feedbacks. Microbial growth and biomass are just part of the many factors influencing these CO₂ production rates.

Additionally, while soils with larger microbial biomass or faster growth rates with warming would also likely have higher rates of CO₂ production, this is not relevant to our particular experiment. As mentioned above in the response about incubation length, because our incubations were short-term, we did not expect microbial biomass to differ over the measurement period. Looking across samples there were clearly differences in microbial biomass, as seen in the PLFA data in Fig. 3d. However, because we compared differences in the parameters extracted from the full temperature response curve, it does not matter if the absolute rates of respiration differed due to different sized microbial pools. Temperature response curves were also scaled to account for these absolute differences when conducting the analyses for Fig. 4.

We agree however, that understanding changes in the temperature response of microbial carbon use efficiency with warming could be an important next step for research in this area. This is now added to the Conclusion (lines 324-325).

4. Smith, T. P. *et al.* Community-level respiration of prokaryotic microbes may rise with global warming. *Nat. Commun.* 10, 1–11 (2019).

5. Alster, C. J., Weller, Z. D. & von Fischer, J. C. A meta-analysis of temperature sensitivity as a microbial trait. *Glob. Chang. Biol.* 24, 4211–4224 (2018).

Are the findings new, or confirmatory?

The authors conclude with “... this research is the first to explicitly quantify the rate of thermal adaptation of soil microbial respiration using a rigorous approach to measuring the full temperature response curve”. Actually, by the authors own appropriate and scholarly literature comparison, this is not the case. Earlier compilations of published data (Bååth, 2018 GCB, MS reference list) and recent large scale surveys (Li et al. 2020; GCB; MS reference list) have both landed on similar estimates for how indices for temperature relationships depend on climate (0.2-0.3 °C change per 1°C warmer MET). Given this, it is awkward to call the findings “unexpected” (l. 307), and certainly to call them “the first” (l. 321). In fact, the estimates fall where they should, based on the few earlier studies that exist, making the finding confirmatory (but naturally still very valuable).

We agree with you that this section was not well worded. As noted in the manuscript text, other soil respiration and microbial growth experiments have observed a <0.5°C change in T_{opt} and T_{min} with a 1°C change in MET. These comparable *in situ* soil-warming studies however, only measured part of the temperature response curve and/or measured across narrower environmental temperature ranges. Several of these studies also focused on microbial growth, which may have similar responses but is not the same as respiration. Because of this, we see our manuscript as most thorough in terms of quantification of the temperature response of respiration over a large environmental temperature gradient. We modified the language to reflect this in the Conclusion and remove reference to “first” (lines 326-328).

Additionally, we also agree that using the term “unexpected” to describe these results is awkward. Rather, we find the mechanisms behind this incommensurate response of thermal adaptation of soil microbial respiration with warming elusive given what we know about the diversity, short generation times, and dispersal of microbial communities. We changed the text in lines 128-129 and removed in the sentence in the Conclusion.

Link between taxonomic composition and temperature? Or pH?

The authors are happy to report a strong link between community diversity measures and environmental temperatures (and microbial temperature adaptations!), as seen in Fig. 3 a, b, c.

However, the authors also show a perfect correlation between pH and MET in their data on bacterial biomass (Fig. 3d). I wonder what would happen if the x-axes of Figs. 3a, b c would be changed from MET to pH? I think there would be a perfect correlation. While one interpretation of this could be that MET was the causal driver, the perfect confounding factor pH may be a more likely factor. Either way, walling (MET) over the other is not appropriate.

This is an interesting point and something we thought a lot about over the course of the experiment and analysis. The pH is likely the most important driver of soil microbial community composition, which is seen in many studies (e.g., ref^{6,7}). The key point we were trying to make in this section is that despite large changes in microbial community composition, which may certainly be driven by pH, shifts in the temperature response were minimal. When we initially conducted the respiration experiment and found such subtle shifts in the temperature response along the temperature gradient, we were concerned that we only found

these results because the microbial communities along the geothermal gradient were potentially very similar. We then decided to look at this more closely and found that the microbial community does in fact change substantially. Therefore, we were able to conclude that the subtle shifts we identified in the temperature response of respiration were not because the microbial communities were all the same. We modified the text to clarify this important point (lines 180-196).

We also altered Fig. 3a-c to emphasize that pH is an important driver of the microbial community and decreases with increasing temperature. To best match up with the rest of the manuscript, we kept the x-axes the same (mean environmental temperature), but changed the colour pattern to correspond to pH. This is now noted in the figure caption (lines 240-242). We also added a new figure to Supplementary Information section 1.1 showing the relationship between temperature and pH for further clarity.

Interestingly, despite pH being an important driver of microbial community composition and strongly correlated with MET along the gradient ($r = -0.91$; $P < 0.0001$), we did not find a strong relationship between pH and the temperature response. Our model selection process by Akaike information criterion (AIC) revealed the most parsimonious model for explaining T_{opt} and T_{inf} includes MET rather than pH. MET was also the most parsimonious model for explaining variation in bacterial diversity, while pH was most predictive of bacterial biomass. These additional analyses are now included in the manuscript (lines 214-217, 405-408, and 441-443; Table S3). It is not best statistical practice to include both MET and pH in the regression because of the collinearity of these terms. Nonetheless, we ran and compared a regression model with both MET and pH. The combination of MET and pH was a better predictor of bacterial diversity measures (Table S3). However, we found that the model with MET alone was still best for explaining variation in T_{opt} and T_{inf} in this study (Table S3), increasing our confidence in MET as a good predictor of the temperature response.

Lastly, while pH is a known driver of microbial community composition, and the microbial community composition may certainly influence the temperature response of respiration (e.g., ref⁶), there is not an obvious mechanistic reason for why pH itself should be driving the temperature response of respiration. Prior studies have also failed to find a strong link between pH and microbial temperature responses for respiration^{5,9}. It is possible that while pH may not directly drive the temperature response of respiration, the interaction between high temperature and low pH causes stress to the microbe indirectly affecting its temperature response. We have added this hypothesis to the manuscript (lines 208-214).

Overall, the likely possibility of pH being a driving mechanism behind changes in microbial community composition and diversity in this study does not invalidate or conflict with the conclusions made about the temperature response of soil microbial respiration.

5. Alster, C. J., Weller, Z. D. & von Fischer, J. C. A meta-analysis of temperature sensitivity as a microbial trait. *Glob. Chang. Biol.* 24, 4211–4224 (2018).
6. Fierer, N. & Jackson, R. B. The diversity and biogeography of soil bacterial communities. *Proc. Natl. Acad. Sci. U. S. A.* 103, 626–631 (2006).
7. Zhou, Z., Wang, C. & Luo, Y. Meta-analysis of the impacts of global change factors on soil microbial diversity and functionality. *Nat. Commun.* 11, (2020).
8. Alster, C. J., Koyama, A., Johnson, N. G., Wallenstein, M. D. & von Fischer, J. C. Temperature sensitivity of soil microbial communities: An application of macromolecular rate theory to microbial respiration. *J. Geophys. Res. Biogeosciences* 121, 1420–1433 (2016).

9. Craine, J. M., Fierer, N. & McLauchlan, K. K. Widespread coupling between the rate and temperature sensitivity of organic matter decay. *Nat. Geosci.* 3, 854–857 (2010).

Other comments

l. 24 “support previous findings that indicate that...”

This has been modified (line 11).

l.25. Acceleratoin of soil C lossess... What about the microbial role in the formation of SOC vis growth?

As explained above, estimating changes in in microbial growth is beyond the scope of this study. We do explicitly state in the abstract however, and everywhere else throughout the manuscript, that these are *potential* changes in limitation and acceleration of soil C losses with temperature. To further emphasize this limitation in the context of microbial growth, we also added a statement to the “Implications for climate change” (lines 276-278).

l. 44-45. This disregards one of the two major fractions microbes use C for; respiration, but also growth (which could form SOC).

As explained above, the focus of this manuscript is on respiration, which most directly contributes to changes in soil carbon-climate warming feedbacks. However, we added a sentence earlier in the paragraph to better define what we are addressing in this study (24-27).

l. 74-77. I think the approach of using a 5 h incubation time at 2-50°C with a surplus of glucose introduces a systematic confounding effect. See general comments.

See the earlier response about why this surplus of glucose was unlikely to cause a systematic confounding effect based on results from our preliminary experiments. This data is now included in Supplementary Information section 1.4.

l. 88. Should changes be differences (they don’t change over time; they vary in the landscape?)

This word is now changed to “differences” (line 78).

l. 132. The problem is not actually “gradual” but that the effect is too small.

This word is now changed to “small” (line 119).

l. 137-140. Good point!

Thank you!

l. 142. Unexpected... But there were earlier reports (you cite them). See general comments

We agree with you. This sentence is now changed (lines 128-129).

l. 150. I am not sure this rationale makes sense. You can find high levels, if MET are sufficiently high. This rationale could explain a cap in the level, but a small effect size per warmer temperature (throughout a wide range of MET)...

We deleted this sentence.

l. 169-182. This is interesting, but it requires a careful deconstruction of your used metric MET, and how it was obtained. Perhaps it would make sense to consider the maximal values of running means (24 h periods)? Should you consider seasonal change? The samples were measured over a long segment of time, which likely varied seasonally in temperature. How was this considered?

In our initial assessment of the data from the geothermal gradient we used the average temperatures from 30-day period prior to the date of sampling to account for temporal differences. However, we did not find a significant difference between using the 30-day average or MET with T_{opt} and T_{inf} ($P > 0.05$). For simplicity, we decided to use MET since this is a common measure of temperature in environmental systems and a relatively simple, gross measure of temperature at annual scales, which was the focus of this manuscript.

Furthermore, by using MET we were able to detect strong links with T_{opt} and T_{inf} with high levels of confidence (i.e., low error surrounding the slopes) indicating that MET is a good predictor of T_{opt} and T_{inf} . Certainly, there may be other, more nuanced relationships with different measures of environmental temperature as well as secondary controls. However, since the focus of our manuscript was on long-term temperature increases, we felt that this measure was most indicative of this change.

We added more details about the calculation of MET to Supplementary Information sections 1.1 and 1.2 and noted that MET is an annual measure in the Main text (line 79). We also added a new Supplementary Information section 1.6 to describe the analyses using the 30-day average temperature prior to sampling and our rationale for choosing the MET metric.

L. 203-204. Perhaps the most likely explanation is soil pH being confounded? This can also be explored with the taxa uncovered? Do the taxa change like they have been shown to along other pH gradients? See general comment

Yes, we agree that pH is likely a major factor driving shifts in microbial community composition. Yet the question remains of why these shifts correspond to only minor changes in the temperature response. While pH does explain variation in T_{opt} and T_{inf} , MET is a stronger predictor of shifts in T_{opt} and T_{inf} (Table S3).

This sentence has been clarified accordingly (lines 195-196) and better explained elsewhere throughout the manuscript (lines 188-190 and 203-217).

L. 214. You can evaluate how likely, based on the taxa and their relative abundances. pH has a very strong phylogenetic signal. Test it!

While quantitative analyses to correct for the compositionality of molecular microbial community data (e.g., differential abundance analysis) is beyond the scope of this study, we provide clear evidence of a strong shift in microbial community composition along the geothermal gradient. We have tested the relationship between the different microbial diversity measures and pH, including two phylogenetic measure of diversity (Faith's PD and Unifrac distance). This information is now included in Table S3. Our models suggest that MET is a better predictor of these phylogenetic metrics than pH. As mentioned earlier, because MET and pH are so highly correlated ($r = -0.91$; $P < 0.0001$), it is not possible to determine the direct effect of pH on the microbial community in this study since issues related to multicollinearity arise. Therefore, we decided to leave this sentence as is, but included more explanatory text to accompany it (lines 208-217).

Fig. 3d. The estimates of bacterial PLFA concentrations seems low. Typical estimates are about 1000 nmol per g SOC. It looks like you are far lower here.

Yes, the PLFA concentrations were quite low, but the pH at the site is low too. Chen et al. 2004¹⁰ note the existence of a threshold of approximately 3.0 on the acidic side for microbial biomass. Wardle 1998¹¹ also suggests that microbial biomass is most stable around a pH of 6.56 with increased turnover of biomass occurring as pH diverges from this value. The pH along the geothermal gradient ranged from 4.26-5.38. Thus, the acidity likely contributed to the low biomass. We added a note of these low biomass values to Supplementary Information section 2.1. Additionally, we reported these estimates as nmol g⁻¹ soil (not nmol g⁻¹ SOC); this may be the main reason that our reported values are significantly lower than you expected.

10. Chen, G. C. & He, Z. Effects of pH on microbial biomass-C and-P in red soils. Red Soils China Their Nature, Manag. Util. 307–314 (2004).

11. Wardle, D. A. Controls of temporal variability of the soil microbial biomass: A global-scale synthesis. Soil Biol. Biochem. 30, 1627–1637 (1998).

l. 246. Actually, the microbes adapt. A consequence of this can be a change of traits, including respiration, but also of other things, like growth. If both change identically – status quo? At least it needs to be acknowledged.

This terminology is now defined/clarified in the Main text (lines 24-27).

l. 262. There WOULD only

The word “would” has been added (line 263).

l. 261-267. This is interesting, but stops short of a little extra sophistication. What about seasonal and diurnal variation in temperatures? These will similarly contribute (asymmetrically) to whether the adaptation will accelerate or slow C loss... What is MET, and why should a metric average be a useful predictor?

As we mentioned earlier, and is now included in the Supplementary material, we did not find a significant difference between using the 30-day average or MET with T_{opt} and T_{inf} ($P > 0.05$). This suggests that temporal variability does not significantly influence the temperature response curve. Additionally, because the focus of the manuscript is on long-term changes in annual temperatures, MET seemed the clearest choice. The utility of this analysis illustrated in Fig. 4 is that these diurnal and temporal variations can be accounted for by changing the location on the y-axis. To clarify this, we added some additional text to this section (lines 254 and 271-272).

l. 302. See general comments

See explanation above for validation of method and improved explanation in the text and supplementary materials.

l. 328-337. Critical information is how estimates of MET were obtained, where they were measured, and how they were compiled. Did different storage time affect them (12 month period of collection)? What were the MET ranges? Seasonal max?

Information on how the MET values were obtained, where they were measured, and how they were compiled is included in Supplementary Information sections 1.1 and 1.2. We did not find differences based on sampling date or incubation date (reflective of storage time) for the geothermal gradient data ($P > 0.05$), which is now included in Supplementary Information section 1.6. We did not do this analysis for the soils from around NZ samples as they were collected as clusters and therefore it was not feasible to tease apart the different overlapping variables. The maximum and minimum average daily temperatures are now reported in Table S1.

1.347-351. I think this introduces a powerful bias (and artefact) with temperature. See general comments

An improved explanation and validation of the methods are now included in the text (lines 356-359, lines 365-377, and Supplementary Information section 1.4).

Reviewer #2:

Quantifying the thermal adaptation of soil microbial respiration is crucial to understanding the potential of carbon cycle feedbacks under climate warming. This manuscript entitled “Quantifying thermal adaptation of soil microbial respiration” by Alster et al. reports a series of temperature response curves of microbial respiration using soils from around New Zealand including from a natural geothermal gradient. The authors estimate the temperature optima (T_{opt}) and inflection point (T_{inf}) of each curve and document that adaptation of microbial respiration occurred at a rate of $0.29^{\circ}\text{C} \pm 0.04$ 1SE for T_{opt} and $0.27^{\circ}\text{C} \pm 0.05$ 1SE for T_{inf} per degree of warming. Finally, the authors conclude that thermal adaptation is demonstrably offset from warming, and calculate the potential for both limitation and acceleration of soil C losses depending on specific soil temperatures. Obviously, measurement of the rate of thermal adaptation of soil microbial respiration is well valuable. Therefore, I think the results of this manuscript is very interesting and meaningful. However, I have several concerns about the experimental design and associated results:

Thank you for the support of our manuscript. We addressed each of your comments below and amended the manuscript accordingly.

1. It is hard to understand how the author combine soil samples from around New Zealand (large spatial scale) and samples from a natural geothermal gradient (small spatial scale) together to analyze excluding scale differences. Obviously, the two batch of samples are from two different ecosystems, farm land and recreation land. Furthermore, although the authors reported that all 48 soil samples were collected over an approximately 12-month period, all samples from farm land were collected in September to December and all samples from recreation land were collected more randomly in the 12-month period. By the way, the authors don't provide detailed distribution of these soil samples around New Zealand. Overall, the sampling strategy in this study lead to too many biases seriously weakening the reliability of the results.

This is a point we considered carefully while planning our experiment and analysing the data. As the reviewer notes, combining data from small and large spatial scales can be problematic since closer data points are more related, creating spatial autocorrelation and potentially biasing results. To account for this issue, we conducted a special type of linear regression incorporating spatial weights called a spatial simultaneous autoregressive (SAR) model that is used to analyse data with unequal spatial distributions. SAR models incorporate spatial autocorrelation using a neighbourhood matrix specifying the relationship between the residuals

at each site and between neighbours¹². We added this explanatory information to the Methods section (lines 401-407).

Due to several logistical constraints, soils were collected at different times. Only 1-3 soils could be measured in the temperature gradient blocks at once and the overall procedure was labour intensive. Therefore, we chose to collect soils on different days in order to conduct the respiration measurements on relatively fresh soils. Some longer gaps in the collection times also occurred due to laboratory closures during the COVID-19 pandemic. However, we found no relationship between sampling or incubation dates and the T_{opt} and T_{inf} for the geothermal gradient data ($P > 0.05$), which instilled confidence that these factors did not alter the temperature response for soil microbial respiration. This information is now added to Supplemental Information section 1.6.

Lastly, sampling points from the geothermal gradient were randomly selected to ensure roughly equal coverage of the different thermal zones. The farmland samples collected from around New Zealand were part of a larger, multi-year project by Manaaki Whenua – Landcare Research surveying soils from around the country. In brief, to select the sites for the New Zealand soil survey a statistical analysis was conducted of historic SOC data to estimate the sample size needed to detect changes in SOC across five land use classes. A sample size of 500 was selected to exceed statistical requirements for analysis. Subsequently, sampling sites were determined using balance acceptance sampling, where sampling sites were selected to maximize geographic distance between sites. These details are described in Hedley et al. 2020¹³. A set number of sites in each land use class were selected for sampling each year, ensuring national representation each year, but there was no set sampling schedule within the year. We received and processed these samples at the University of Waikato in the order that they arrived. Due to the length this nation-wide soil sampling campaign and intensity of the incubation measurements we conducted, not all of the samples collected by Manaaki Whenua – Landcare Research were included in this experiment. Once a sufficient sample size had been obtained for our analysis, as determined by a post-hoc power analysis (power > 0.99), we ceased data collection for this experiment. This information is now included in Supplementary information sections 1.1 and 1.2.

Ultimately, we felt that the combination of the very different soil types and even differences in seasonality were a strength of this study as the relationship between environmental temperature and T_{opt} and T_{inf} held with such high confidence despite the potential for noise from these confounding variables.

12. F. Dormann, C. *et al.* Methods to account for spatial autocorrelation in the analysis of species distributional data: A review. *Ecography (Cop.)*. 30, 609–628 (2007).

13. Hedley, C. *et al.* Reference manual for implementation of a soil organic carbon monitoring programme: contract report LC3773 for Ministry for Primary Industries. (2020).

2. Based on the description in the methods, the authors collected new 28 soil samples from 14 points along the geothermal gradient to perform microbial community analyses rather than using the same samples as measurement of the temperature response of soil microbial respiration. Why? The authors don't report the sampling time and specific locations. If these samples don't match the measurement of the temperature response of soil microbial respiration, it is not reasonable to explain thermal adaptation of soil microbial respiration by using the data of PLFA or DNA analysis in this study.

The main finding from our manuscript was that thermal adaptation of soil microbial respiration occurred a very modest rate ($<0.3^{\circ}\text{C}$ shift in the temperature response curve per 1°C increase in mean soil temperature). When we initially conducted the respiration experiment and found such subtle shifts in the temperature response along the temperature gradient, we were concerned that we only found these results because the microbial communities along the geothermal gradient were potentially very similar. We then decided to look at this more closely and found that the microbial community does in fact change substantially along the gradient. Therefore, we were able to conclude that the subtle shifts we identified in the temperature response were not because the microbial communities were all the same. A specific match between the temperature response of soil microbial respiration and the PLFA/DNA data was not necessary to make this observation and can be inferred from the regression analyses. The section entitled “The role of the microbial community” has been extensively edited and hopefully now clarifies this misunderstanding.

From a logistical standpoint, there were a few reasons why a slightly different design was chosen for the microbial community characterisation compared with the respiration analyses. Soils for the PLFA and DNA analyses were collected after most of the respiration measurements were completed. Many soil cores (>8) were collected to conduct the respiration analyses. Because this site is located in a public area, we were required to “backfill” the holes with soil that were made from the soil coring. To avoid accidentally re-sampling these backfilled holes, which may have altered our results, we purposefully took samples from other locations along the gradient. We also decided to take two samples from each location (one from the left side and one from the right side of the width of the gradient). For the respiration measurements, we sampled across a 1 m perpendicular line (see Supplementary Information section 1.1). Because so much less soil was needed for the microbial community characterisation and we did not want to needlessly collect excess soil from this unique site, we took these two “replicate” samples to check for potential variability along the width of the gradient. Additionally, taking these two samples at each location ensured an additional sample in the event laboratory difficulties (i.e., low sequence reads). A summary of these details is now added to Supplementary Information section 2.

The sampling dates for the PLFA and DNA data are reported in Supplementary Information section 2. These samples for the PLFA and DNA data were collected on separate days based on researcher availability to process the samples, minimizing the time between sample collection and processing times. The specific sampling locations along the gradient are listed in Table S4. However, for further clarity we also added this information to Supplementary Information section 2.

3. It seems the author estimate the relationship between mean environmental soil temperature and the T_{opt} and T_{inf} of soil microbial respiration by using annual mean environmental soil temperature. If so, the T_{opt} and T_{inf} of soil microbial respiration estimated by soil samples from different month can't represent the mean value across the whole year. It is not reasonable to quantify relationship between MET and the T_{opt} and T_{inf} .

We understand your hesitation here. In our initial assessment of the data, we used the average temperatures for the 30-days prior to soil sampling. However, we found no difference between using this metric versus MET over the year ($P > 0.05$). For simplicity, we decided to use MET since this is a common measure of temperature in environmental systems and a relatively simple, gross measure of temperature at annual scales. Furthermore, the focus of our manuscript was on long-term temperature increases and MET captures this well. This information is now included in Supplementary Information section 1.6. That we observe such a strong correlation between MET and T_{opt} and T_{inf} despite sampling at different time of year

suggests that sampling month was not that critical for observing the temperature response of respiration.

4. In the result part about Thermal adaptation of soil microbial respiration, the authors perform analysis by using both samples from farm land around New Zealand and from the natural geothermal temperature gradient. However, only samples from the geothermal gradient were further analysis in the role of the microbial community. Strictly, it is not acceptable to explain the overall thermal adaptation obtained in this study by only using a part of samples.

We understand why this was confusing – thank you for bringing it to our attention. Our intention was to characterise the microbial community from along the geothermal gradient to check that the communities were in fact distinct. Since the other samples were taken from such separate location and soil types, we felt it was safe to assume that these microbial communities were in fact different^{14,15}. We added an explanation of our rationale to the manuscript (lines 180-183).

Additionally, we decided to break down the geothermal gradient data separately to minimize confounding variables that were introduced in the full country sampling. The fact that the results were so similar for the rate of change in T_{opt} and T_{inf} in both the full dataset and the geothermal gradient, strengthens our findings that these results are not simply an artefact of many variables obscuring a true relationship. We revised lines 171-186 to explain this more clearly.

14. Ettema, C. H. & Wardle, D. A. Spatial soil ecology. Trends Ecol. Evol. 17, 177–183 (2002).

15. Constancias, F. *et al.* Mapping and determinism of soil microbial community distribution across an agricultural landscape. Microbiologyopen 4, 505–517 (2015).

5. Line 122-123: the authors don't show any results about T_{min} in the study. I don't understand why the T_{min} should be discussed here.

Few studies quantify the rate of thermal adaptation of soil microbial communities. Therefore, we decided to include all relevant studies on this topic, including ones measured in different ways (i.e., T_{min}). While certainly not a perfect parallel to T_{opt} or T_{inf} , the rate of change for T_{min} was similar and thus seemed important to include. In Box 1 there is an explanation of T_{min} and comparisons to parameters from MMRT. We have now added a note to “see Box 1” for further context in this sentence (line 110).

6. Line 218-219: this sentence is not rigorous. Definitely temperature is important to control respiration rates, but the other factors, such as soil moisture, also strongly control respiration rates. Therefore, definite evidence should be provided in this study to document your viewpoint.

We agree. This sentence has been removed and the paragraph has been modified to better illustrate the relevant interactions we observed between pH and the soil temperature response and soil microbial communities.

Additionally, while soil moisture would certainly control the respiration rates, it is unlikely that soil moisture would change the shape of the temperature response curve². We changed the language in this paragraph to clarify that we are discussing the temperature response curve of respiration, not absolute values of respiration (lines 203-206).

2. Schipper, L. A. *et al.* Shifts in temperature response of soil respiration between adjacent irrigated and non-irrigated grazed pastures. *Agric. Ecosyst. Environ.* 285, 106620 (2019).

7. Please provide links for the raw data of amplicon sequences.

Raw sequence data is now provided in Table S5.

REVIEWERS' COMMENTS

Reviewer #1 (Remarks to the Author):

Review of NCOMMS-23-07213A

The initial draft of this MS was already a well written account of a competent set of experiments. In its revised form, the authors have thoroughly considered and responded to the comments raised by the two reviewers. This has led to several appropriate changes to the MS. For instance, the linkage between taxonomic composition and temperature variables is now far better nuanced, also acknowledging the (likely) role of soil pH differences. However, the revision has also substantially changed the case for novelty. In the authors own assessment (copied from the responses to reviewers): “the main finding from our manuscript was that thermal adaptation of soil microbial respiration occurred at a very modest rate (<0.3°C shift in the temperature response curve per 1°C increase in mean soil temperature)”.

As highlighted in the first round of review, there are several previous reports in recent years that have reported similar findings, even landing on an average effect size very similar to that arrived at in this study (0.2-0.3°C per 1°C difference in MAT). This is now more carefully considered in the revised discussion (l. 105-115), but is still neglected in the authors formulation of the knowledge gap and scope of their study (l. 32-35). The authors still imply critical knowledge gaps, and lacking data to address them in... This is later contradicted by their own literature survey (l. 105-115), making the MS problematically contradictory...

In the response letter, the authors highlight that most of those previous reports on the rate of adaptation of thermal adaptation were based on changes in temperature relationships of microbial growth rates. This is partly justified. But not fully. When temperature relationships for respiration (Bååth, 2018, Li et al. 2020; cited in MS) are considered across various ecosystems, in response to warming, and across broad environmental gradients of temperature, estimates of the rate of change are 0.2-0.3°C change per 1°C change in MAT... This is just what the authors highlight as their central finding! While this validation is good

scientific progress, this makes the findings confirmatory, rather than novel. For many journals, this is not an issue, but I think novelty is an important criteria for publication in an interdisciplinary journal that premier novelty, such as Nat Comm, which is also why I think the introduction still maintains the illusory knowledge gaps and open questions (l. 32-35) that do not actually exist (rather, they are not open gaps without information). Rather, a more appropriate reformulation of the introduction and aim would be the hypothesis that “the rate of change of temperature relationships for respiration should be 0.2-0.3°C change per 1°C difference in MAT... While making the MS far less novel, it would also be a very precise hypothesis, that is well anchored in recent literature.

The authors also argue that previous assessments were less thorough in terms of MAT ranges considered, or in terms of the breadth of the temperature relationships estimated. This is, again, is not really justified. MAT ranges from Li et al. were from -6°C to +25, and respiration determined between 4-28°C. They included 298 samples! Another still more recent study included temperature relationships for respiration (0-45°C) along with fungal and bacterial growth (same ranges) and 72 sites across Europe from -3 – 18 °C of MATs (Cruz Paredes et al. 2023, <https://journals.asm.org/doi/10.1128/aem.02090-22>). Both of these large scale surveys found similar ranges of estimates for rates of change of temperature relationships, from 0.1°C to 0.3 °C.

None of these concerns call the quality of the submitted MS into question. Rather, they illustrate that the findings are expected, and actually confirmatory, rather than novel. What should be called into question is the authors’ current framing of their finding as novel. The specific rate of change could have been (should have been?) anticipated to be far less than 1°C per 1°C difference in MAT, which still is the central emphasis for novelty claimed by the authors...

What is novel is the authors numerical exercises, where they draw on their findings to infer how rates of respiration are affected, and how this varies with the current environmental temperature. However, this exercise is still incomplete. If decomposer microbes indeed adapt to changes in temperature such that their temperature relationships for respiration shift (as the authors find, and which is consistent with several recent reports), then it is

likely that the temperature relationships for microbial processes that give rise to soil C storage (namely microbial growth) also shift. The recent reports that have found similar rates of temperature curve shifts to those reported by the authors have actually shown this, meaning that the literature is already cited and known by the authors. If the authors target is to understand (or scratch the surface of understanding) of how warming will affect the temperature forcing of microbial processes that translates into changes in soil C stocks, this is an odd thing to disregard, even as a back of the envelope calculation. The authors are more carefully restricting their discussion to “changes in potential respiration” (l. 267-281), and argue that other microbial responses are “our of scope” (the authors’ answer in their response letter). However, the authors still transition into a discussion of soil C losses (l. 283-287), which require a consideration also of if inputs change (consider the microbial carbon pump, Liang et al. 2017; there input minus output equal changes in soil C stocks). If the authors are interested in how microbial adaptation to warming will affect soil C stocks, responses in both input and output are needed (l.283- 293), and results have been reported providing estimates for changes in the temperature relationships for both (see comments above).

Other comments

The authors appropriately address the methodological concerns raised in the initial round of review concerning basal vs substrate-saturated respiration. However, why not include both sets (since the authors clearly have both, given that the presented curves are the difference between them) in addition to the derived difference curves that actually are used as the authors primary data. I think this will facilitate effective comparison with other studies, that will be valuable to the field.

Reviewer #2 (Remarks to the Author):

In this revision, Dr. Alster and co-authors have done a good job at addressing the questions and comments previously raised by the reviewers including myself. The authors have clarified the methods of data analysis and text that were previously vague, which is appreciated. The authors have also convincingly shown that the results observed are novel. Thanks a lot for your efforts, I'm satisfied with the current modification. I look forward to seeing this study in its final published version.

Both reviewers were very positive on the technical quality of our experiment and write-up of the manuscript. Reviewer #1 raised some questions about the novelty of our findings. We respectfully disagree with Reviewer #1's opinion. There are several key aspects of our work that distinguishes it from other, seemingly similar studies. These include:

- 1) Precision of model estimates allowing for a more reliable analysis of the data and resulting robust conclusions**
- 2) Separation of substrate availability from microbial respiration**
- 3) Dispersal of microbiological communities is unlikely to be the reason for muted adaptation as sample sites along the geothermal gradient were only meters apart and had likely been under different temperatures for decades.**
- 4) Generation of new hypotheses crucial for predicting soil microbial adaptation under climate warming, providing important conceptual advances**

Below, we provide a more detailed response to the reviewer comments (in bold).

Reviewer #1 (Remarks to the Author):

Review of NCOMMS-23-07213A

The initial draft of this MS was already a well written account of a competent set of experiments. In its revised form, the authors have thoroughly considered and responded to the comments raised by the two reviewers. This has led to several appropriate changes to the MS. For instance, the linkage between taxonomic composition and temperature variables is now far better nuanced, also acknowledging the (likely) role of soil pH differences. However, the revision has also substantially changed the case for novelty. In the authors own assessment (copied from the responses to reviewers): “the main finding from our manuscript was that thermal adaptation of soil microbial respiration occurred at a very modest rate ($<0.3^{\circ}\text{C}$ shift in the temperature response curve per 1°C increase in mean soil temperature)”.

As highlighted in the first round of review, there are several previous reports in recent years that have reported similar findings, even landing on an average effect size very similar to that arrived at in this study ($0.2\text{-}0.3^{\circ}\text{C}$ per 1°C difference in MAT). This is now more carefully considered in the revised discussion (l. 105-115), but is still neglected in the authors formulation of the knowledge gap and scope of their study (l. 32-35). The authors still imply critical knowledge gaps, and lacking data to address them in... This is later contradicted by their own literature survey (l. 105-115), making the MS problematically contradictory...

In the response letter, the authors highlight that most of those previous reports on the rate of adaptation of thermal adaptation were based on changes in temperature relationships of microbial growth rates. This is partly justified. But not fully. When temperature relationships for respiration (Bååth, 2018, Li et al. 2020; cited in MS) are considered across various ecosystems, in response to warming, and across broad environmental gradients of temperature, estimates of the rate of change are $0.2\text{-}0.3^{\circ}\text{C}$ change per 1°C change in MAT... This is just what the authors highlight as their central finding! While this validation is good scientific progress, this makes the findings confirmatory, rather than novel. For many journals, this is not an issue, but I think novelty is an important criteria for publication in an interdisciplinary journal that premier novelty, such as Nat Comm, which is also why I think the introduction still maintains the illusory knowledge gaps and open questions (l. 32-35) that do not actually exist (rather, they are not open gaps without information). Rather, a more appropriate reformulation of the introduction and aim would be the hypothesis that “the rate of change of temperature relationships for respiration should be $0.2\text{-}0.3^{\circ}\text{C}$ change per 1°C difference in MAT...

While making the MS far less novel, it would also be a very precise hypothesis, that is well anchored in recent literature.

The authors also argue that previous assessments were less thorough in terms of MAT ranges considered, or in terms of the breadth of the temperature relationships estimated. This is, again, is not really justified. MAT ranges from Li et al. were from -6°C to $+25$, and respiration determined between $4\text{--}28^{\circ}\text{C}$. They included 298 samples! Another still more recent study included temperature relationships for respiration ($0\text{--}45^{\circ}\text{C}$) along with fungal and bacterial growth (same ranges) and 72 sites across Europe from $-3\text{--}18^{\circ}\text{C}$ of MATs (Cruz Paredes et al. 2023, <https://journals.asm.org/doi/10.1128/aem.02090-22>). Both of these large scale surveys found similar ranges of estimates for rates of change of temperature relationships, from 0.1°C to 0.3°C .

None of these concerns call the quality of the submitted MS into question. Rather, they illustrate that the findings are expected, and actually confirmatory, rather than novel. What should be called into question is the authors' current framing of their finding as novel. The specific rate of change could have been (should have been?) anticipated to be far less than 1°C per 1°C difference in MAT, which still is the central emphasis for novelty claimed by the authors...

It is interesting that the estimates for thermal adaptation of microbial respiration found in our study mirror those from some other studies; however, we disagree that this is necessarily expected or takes away from the importance of our findings. Not all similar studies find confirmatory results (e.g., Moinet et al. 2021). Additionally, despite similar results to a few studies, the methodological differences do not make the findings as 'expected' as the reviewer suggests. We will try to avoid being overly pedantic in our explanation here, but despite being excellent studies, Bååth 2018, Li et al. 2020, and Cruz Paredes et al. 2023 (the studies highlighted by the reviewer as being most comparable to ours) do not offer the same level of precision or understanding of the temperature response. This is for a few key reasons:

- 1. All three of these 'comparable' studies estimate T_{\min} for the temperature response metric rather than measuring the full temperature response curve (i.e., plotting a line to one part of the temperature response curve and specifically excludes temperatures near T_{opt}). While arguably useful as a gross proxy for thermal adaptation, this metric does not fully capture the nuance of changes in the temperature response. A 0.2°C increase in T_{\min} per 1°C increase in mean soil temperature, for example, does not necessarily correspond to a 0.2°C change in T_{opt} , T_{inf} , or other features of the temperature response curve. Furthermore, unlike T_{opt} and T_{inf} , T_{\min} is an extrapolated, theoretical measure of the temperature response and should be treated as such. The fact that T_{\min} , T_{opt} , and T_{inf} all shift at relatively the same rate with increasing temperature suggests that the shape of the temperature response curve stays relatively constant rather than changing shape as temperature increases. This result is also unexpected based on the enzymatic literature where the shape of the temperature response curve narrows with cooler temperatures to compensate for declining activity at lower temperatures (e.g. Struvay and Feller 2012).**
- 2. We estimated the temperature response of microbial respiration with standardized substrate availability, which is a marked difference from the other studies mentioned. The differences in substrate availability between samples in the other studies could potentially mask the true rate of adaptation. We see in these other studies that the error around the slope estimates are higher than ours, which perhaps reflects the noise presented by this confounding factor (among other issues mentioned in point 1). It is likely that by including variation in substrate availability, we would not have observed the same rate of change (Alster et al. 2022), thus making our observations unexpected.**

3. We can exclude limited dispersal of microorganisms for slower than expected thermal adaptation because sample sites along the geothermal gradient were only meters apart and had likely been under different temperatures for decades. Soil mixing would have allowed for plenty of opportunities for migration and replacement of existing communities.
4. Our work provides a conceptual advance beyond the ‘enhancement’ versus ‘compensatory’ hypotheses of thermal adaptation of soil microbial respiration, which has dominated the literature over the past few years (e.g., Bradford et al. 2019, Ye et al. 2020, Zhang et al. 2023), by considering the rate of thermal adaptation simultaneous to the instantaneous temperature. This important body of literature and advancement that our work provides is ignored in the reviewer’s critique.

Because of these reasons, we disagree that our manuscript is ‘problematically contradictory’ and that the findings lack novelty. We summarize these points in the second paragraph in the Introduction, Box 1, and the ‘Implications for climate change’ section.

References:

- Moinet, G. Y. K. et al. Soil microbial sensitivity to temperature remains unchanged despite community compositional shifts along geothermal gradients. *Glob. Chang. Biol.* 27, 6217–6231 (2021).
- Bååth, E. Temperature sensitivity of soil microbial activity modeled by the square root equation as a unifying model to differentiate between direct temperature effects and microbial community adaptation. *Glob. Chang. Biol.* 24, 2850–2861 (2018).
- Li, J., Bååth, E., Pei, J., Fang, C. & Nie, M. Temperature adaptation of soil microbial respiration in alpine, boreal and tropical soils: An application of the square root (Ratkowsky) model. *Glob. Chang. Biol.* (2020) doi:10.1111/gcb.15476.
- Cruz-Paredes, C., Tájmel, D. & Rousk, J. Variation in Temperature Dependences across Europe Reveals the Climate Sensitivity of Soil Microbial Decomposers. *Appl. Environ. Microbiol.* 89, e0209022 (2023).
- Struvay, C. & Feller, G. Optimization to low temperature activity in psychrophilic enzymes. *Int. J. Mol. Sci.* 13, 11643–11665 (2012).
- Alster, C. J., Robinson, J. M., Arcus, V. L. & Schipper, L. A. Assessing thermal acclimation of soil microbial respiration using macromolecular rate theory. *Biogeochemistry* 158, 131–141 (2022).
- Bradford, M. A. et al. Cross-biome patterns in soil microbial respiration predictable from evolutionary theory on thermal adaptation. *Nat. Ecol. Evol.* 3, 223–231 | (2019).
- Ye, J. S., Bradford, M. A., Maestre, F. T., Li, F. M. & García-Palacios, P. Compensatory Thermal Adaptation of Soil Microbial Respiration Rates in Global Croplands. *Global Biogeochem. Cycles* 34, (2020).
- Zhang, Y. et al. Temperature fluctuation promotes the thermal adaptation of soil microbial respiration. *Nat. Ecol. Evol.* 7, 205–213 (2023).

What is novel is the authors numerical exercises, where they draw on their findings to infer how rates of respiration are affected, and how this varies with the current environmental temperature. However, this exercise is still incomplete. If decomposer microbes indeed adapt to changes in temperature such that their temperature relationships for respiration shift (as the authors find, and which is consistent with several recent reports), then it is likely that the temperature relationships for microbial processes that give rise to soil C storage (namely microbial growth) also shift. The recent reports that have found similar rates of temperature curve shifts to those reported by the authors have actually shown this, meaning that the literature is already cited and known by the authors. If the authors target is to understand (or scratch the surface of understanding) of how warming will affect

the temperature forcing of microbial processes that translates into changes in soil C stocks, this is an odd thing to disregard, even as a back of the envelope calculation. The authors are more carefully restricting their discussion to “changes in potential respiration” (l. 267-281), and argue that other microbial responses are “out of scope” (the authors’ answer in their response letter). However, the authors still transition into a discussion of soil C losses (l. 283-287), which require a consideration also of if inputs change (consider the microbial carbon pump, Liang et al. 2017; there input minus output equal changes in soil C stocks). If the authors are interested in how microbial adaptation to warming will affect soil C stocks, responses in both input and output are needed (l.283- 293), and results have been reported providing estimates for changes in the temperature relationships for both (see comments above).

We agree that this numerical aspect of our study provides an important conceptual advance on the topic of thermal adaptation of soil microbial adaptation. This type of analysis is made possible with the empirical data we obtained and further differentiates it from similar studies in this field. However, this detailed level of understanding of microbial thermal responses is just one component in predicting if total carbon stocks will increase or decrease with warming. Before we can determine if total C stocks will increase or decrease with warming, we would need to couple the microbial changes with plant responses to warming (i.e., plant photosynthesis, respiration, and belowground inputs). While assumptions could be made about these plant responses to warming to conduct a ‘back of the envelope’ calculation, it is highly likely that these tenuous assumptions would result in misleading predictions causing more harm than help.

However, we agree that it is important to make note of these caveats of concurrent changes in microbial biomass and substrate availability (strongly influenced by plants), which will also change with warming. For clarity, we modified lines 257-259: “Note, that it is potential respiration rate, because these experiments were conducted without substrate limitation and do not account for potential changes in substrate supply or microbial biomass, which may also adapt differentially to warming.”

Other comments

The authors appropriately address the methodological concerns raised in the initial round of review concerning basal vs substrate-saturated respiration. However, why not include both sets (since the authors clearly have both, given that the presented curves are the difference between them) in addition to the derived difference curves that actually are used as the authors primary data. I think this will facilitate effective comparison with other studies, that will be valuable to the field.

The data for these curves is provided in Table S7.

Reviewer #2 (Remarks to the Author):

In this revision, Dr. Alster and co-authors have done a good job at addressing the questions and comments previously raised by the reviewers including myself. The authors have clarified the methods of data analysis and text that were previously vague, which is appreciated. The authors have also convincingly shown that the results observed are novel. Thanks a lot for your efforts, I’m satisfied with the current modification. I look forward to seeing this study in its final published version.

Thank you for the support of our manuscript!